# IL36 is a critical upstream amplifier of neutrophilic lung inflammation in mice

Carolin K. Koss[1,2], Christian T. Wohnhaas[1,2], Jonathan R. Baker[3], Cornelia Tilp[1], Michèl Przibilla[1], Carmen Lerner[1], Silvia Frey[1], Martina Keck[1], Cara M. M. Williams[1,5], Daniel Peter[1], Meera Ramanujam[4], Jay Fine[4], Florian Gantner[1,2], Matthew Thomas[1], Peter J. Barnes[3], Louise E. Donnelly[3] & Karim C. El Kasmi [1 ✉]

IL-36, which belongs to the IL-1 superfamily, is increasingly linked to neutrophilic inflammation. Here, we combined in vivo and in vitro approaches using primary mouse and human cells, as well as, acute and chronic mouse models of lung inflammation to provide mechanistic insight into the intercellular signaling pathways and mechanisms through which IL-36 promotes lung inflammation. IL-36 receptor deficient mice exposed to cigarette smoke or cigarette smoke and H1N1 influenza virus had attenuated lung inflammation compared with wild-type controls. We identified neutrophils as a source of IL-36 and show that IL-36 is a key upstream amplifier of lung inflammation by promoting activation of neutrophils, macrophages and fibroblasts through cooperation with GM-CSF and the viral mimic poly(I:C). Our data implicate IL-36, independent of other IL-1 family members, as a key upstream amplifier of neutrophilic lung inflammation, providing a rationale for targeting IL-36 to improve treatment of a variety of neutrophilic lung diseases.

[1] Boehringer Ingelheim Pharma GmbH & Co KG, Biberach, Germany. [2] Department of Biology, University of Konstanz, Konstanz, Germany. [3] Airway Disease, National Heart and Lung Institute, Imperial College London, London, UK. [4] Boehringer Ingelheim Pharmaceuticals Inc., Ridgefield, CT, USA. [5] Present address: WRDM, Inflammation and Immunology Research Unit, Pfizer, Cambridge, MA, USA. ✉email: karim_christian.el_kasmi@boehringer-ingelheim.com

The recently described interleukin (IL)-1 family cytokines IL-36α, IL-36β, and IL-36γ[1–3] are emerging as contributors to acute and chronic tissue inflammation in human disease, particularly in the neutrophilic skin disease psoriasis[4–6]. In addition, experimental animal models of skin inflammation, arthritis, and intestinal inflammation further implicate IL-36 cytokines as important inflammatory mediators[7–11]. In the lung, IL-36 has been suggested to be important in the pathogenesis of experimental bacterial and viral pneumonia in mice[7,12]. IL-36 cytokines signal via a heterodimeric receptor comprised of an IL-36 receptor (IL-36R) chain and the IL-1 receptor accessory protein (IL-1RAP) that is also shared with the IL-1 and IL-33 receptors[13,14]. IL-36 cytokines can be expressed by skin epithelial cells[9,11], which correlates with increased mRNA and protein concentrations of IL-36 cytokines in human psoriatic skin lesions and experimental models of psoriasis-like diseases in mice, including acanthosis and hyperkeratosis[4,5,8–11,15]. A common finding in diseases where IL-36 cytokines play a role is the presence of neutrophils[8]. Like other IL-1 family cytokines, IL-36 cytokines are produced as precursor proteins whose bioactivity is 1000-fold increased after proteolytic processing[16,17]. Typically, activation of IL-36 cytokines is associated with increased expression of multiple proteases released by neutrophils, such as neutrophil-derived cathepsin G, elastase, and proteinase-3[18]. Consistent with the link between IL-36 cytokines and neutrophils, IL-36 cytokines have been strongly linked to the pathogenesis of generalized pustular psoriasis (GPP) and hidradenitis suppurativa, diseases in which neutrophils are a hallmark feature[19]. Importantly, blocking IL-36 receptor with a monoclonal antibody markedly reversed the skin lesions in human subjects with GPP[20]. IL-36 signaling is also activated in intestinal inflammation such as inflammatory bowel disease and experimental colitis[21,22]. Furthermore, IL-36 also plays an important role in the joint synovium of patients with rheumatoid arthritis[23,24]. These studies have extended IL-36 and IL-36R expression to include fibroblasts and macrophages[25–28], and some reports have also suggested IL-36 expression by lung epithelial cells[29,30].

Thus, there is a rationale for considering IL-36 cytokines as important contributors to the pathogenesis of lung diseases, specifically those that are characterized by the accumulation of neutrophils, such as severe non-T2 asthma, chronic obstructive pulmonary disease (COPD), acute respiratory distress syndrome, and cystic fibrosis[31]. Associative data have linked IL-36 cytokine(s) to neutrophilic inflammation in multiple diseased tissues, yet the mechanisms by which IL-36 drives pathology remain elusive. Here, we used both wild-type (WT) and knockout (KO) in vitro and in vivo murine experimental approaches together with primary human cells to deconvolute the network of stimuli and responses that orchestrate neutrophilic lung disease. Here, we provide the mechanistic rationale by which targeting IL-36 may alleviate our most burdensome respiratory diseases.

## Results

**IL-36 in neutrophilic lung inflammation.** We exposed WT and IL-36 receptor KO mice ($Il36r^{-/-}$) to cigarette smoke (CS) for 3 weeks. Exposure of WT mice to CS resulted in significantly increased bronchoalveolar lavage (BAL) neutrophil numbers ($4.5 \times 10^5$ neutrophils/mL). In contrast, alveolar macrophage (AM) numbers did not change after exposure to CS. In $Il36r^{-/-}$ CS-exposed mice ($1.7 \times 10^5$ neutrophils/mL), we observed a 62% reduction in BAL neutrophil numbers relative to WT mice. In addition, myeloperoxidase (as an indicator of neutrophil activation) concentrations in the BAL of $Il36r^{-/-}$ CS-exposed mice were significantly reduced (to concentrations observed in mice exposed to room air (RA)) relative to CS-exposed WT mice

(Fig. 1a). These experiments indicated that IL-36 receptor (encoded by $Il1rl2$, herein designated as $Il36r$) signaling was an upstream driver of neutrophil recruitment and activation in mice exposed to CS smoke.

To further determine whether IL-36 cytokines acted as upstream drivers of neutrophil recruitment in the lung, we instilled IL-36γ (as one representative IL36 family cytokine previously described in the literature as a potent pro-inflammatory stimulus)[32] intratracheally (i.t.) into naive mice for 4 h (in order to examine early upstream events before the emergence of secondary effects). In contrast to AM numbers, IL-36γ significantly increased BAL neutrophil numbers (Fig. 1b) to amounts comparable to those observed in CS-exposed mice (Fig. 1a). Additionally, C-X-C chemokine ligand 1 (CXCL1; a key neutrophil chemoattractant protein) concentrations were increased in the BAL relative to phosphate-buffered saline (PBS)-exposed mice (Fig. 1c). IL-36γ also significantly increased concentrations of IL-1α, IL-1β, and granulocyte macrophages colony-stimulating factor (GM-CSF; cytokines typically generated by neutrophils, macrophages, or the epithelium[33]) in the BAL relative to PBS-exposed mice (Fig. 1d). In addition, relative $Cxcl1$ and $Il6$ mRNA amounts were increased within the cell pellet recovered from the BAL relative to PBS exposure (Suppl. Fig. 1a).

We next determined whether IL-36 cytokines directly promoted pro-inflammatory activation of AMs and neutrophils. Murine AMs obtained from BAL of naive mice were cultured in the presence of GM-CSF (to maintain normal AM function[34,35]) and exposed to IL-36 cytokines for 24 h, after which $Il1a$ and $Il1b$ mRNA expression was determined. Both $Il1a$ (19-fold) and $Il1b$ (2-fold) expression was significantly increased relative to that in unstimulated AMs (Fig. 1e). Moreover, relative to unstimulated AMs, IL-36 cytokine-stimulated AMs also had significantly increased mRNA expression for $Il36g$ (4-fold) and $Cxcl1$ (30-fold), indicating AMs as a source of CXCL1 and IL-36γ in the alveolar compartment. Importantly, a combination of IL-1α and IL-1β did not induce transcription of $Il1a$, $Il1b$, or $Cxcl1$ and only mildly increased $Il-36g$ in AMs (Fig. 1e). We also examined the effect of other lung-associated inflammatory mediators by exposing AMs to lipopolysaccharide (LPS) or its canonical downstream cytokine tumor necrosis factor (TNF)-α. Exposing AMs to TNF-α did not stimulate mRNA expression for $Il36g$ or $Cxcl1$ in AMs (Suppl. Fig. 1b), while the fold induction of mRNA encoding $Cxcl1$, $Il36g$, $Il1b$, $Il1a$, and $Il6$ after stimulating AMs with LPS was similar to that observed in response to IL-36 cytokines, indicating IL-36 cytokines as a potent pro-inflammatory stimulus on AMs (Suppl. Fig. 1c).

Considering that AMs were cultured in GM-CSF, we stimulated bone marrow-derived neutrophils from naive mice in vitro with IL-36γ alone or in combination with GM-CSF. Combining GM-CSF with IL-36γ resulted in significantly increased mRNA expression for $Il36g$ (4-fold), $Cxcl1$ (10-fold), $Il1a$ (600-fold), and $Il1b$ (24-fold) relative to IL-36γ alone or GM-CSF alone when compared to unstimulated neutrophils (Fig. 1f). GM-CSF was capable of stimulating $Il36g$ expression in mouse neutrophils, suggesting a further mechanism by which IL-36γ can be induced. As was observed for AMs, IL-1α and IL-1β stimulation of neutrophils failed to induce gene expression for these cytokines, even in combination with GM-CSF (Suppl. Fig. 1d). These findings prompted us to determine whether GM-CSF stimulation increased mRNA expression of the $Il36r$ in neutrophils. Indeed, GM-CSF exposure resulted in an ~10-fold increase in $Il36r$ mRNA expression in neutrophils, while no change in the expression of $Il1r1$ was observed (Fig. 1g). GM-CSF also significantly induced the mRNA expression of the IL-1 receptor antagonist ($Il1rn$), whereas the mRNA expression of the IL-36 receptor antagonist ($Il36rn$) remained below the detection

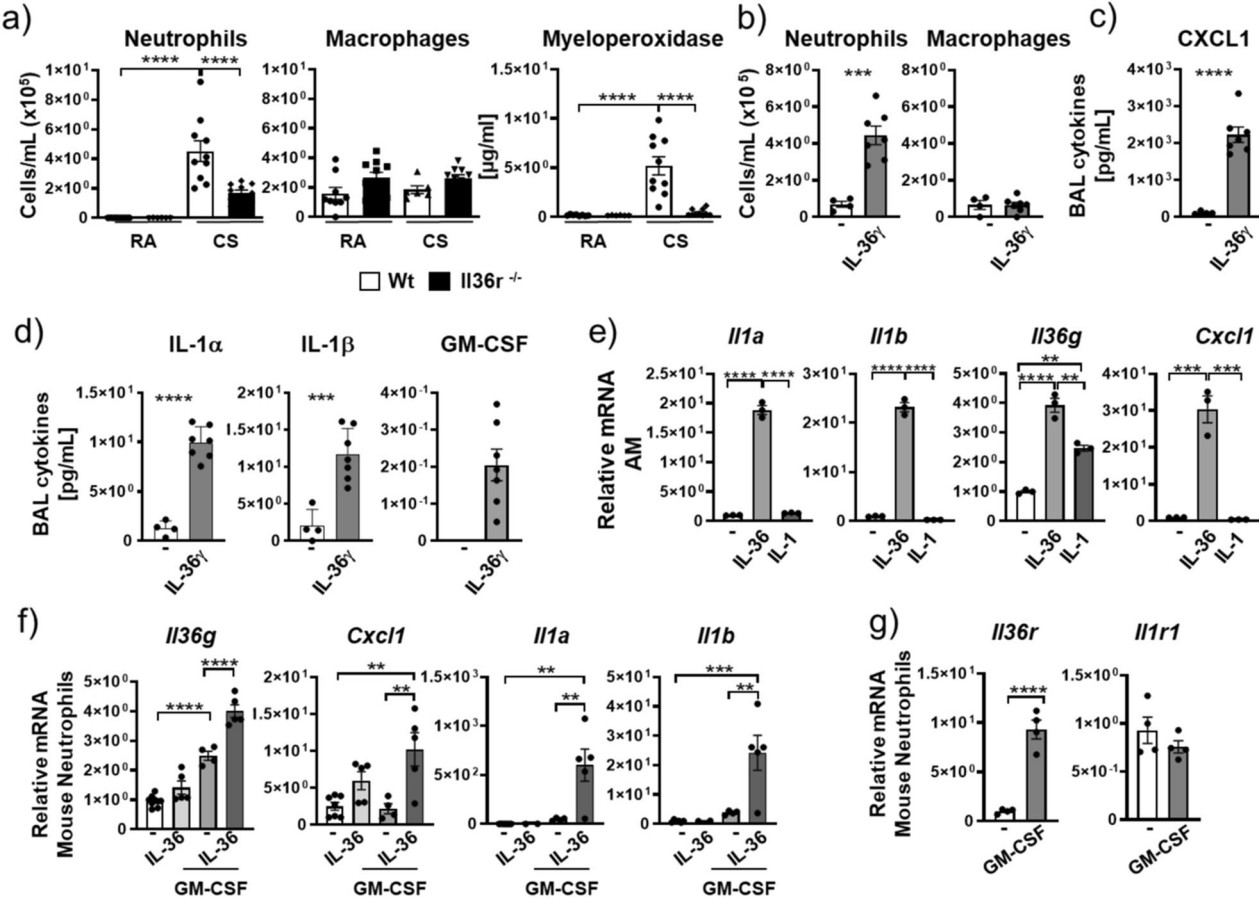

**Fig. 1 IL-36γ is an upstream inflammatory driver in mouse neutrophils and alveolar macrophages. a** Neutrophil and macrophage counts and myeloperoxidase concentration in the bronchoalveolar lavage fluid (BALF) from room air (RA) exposed (WT $n = 9$; $Il36r^{-/-}$ $n = 6$) and 3-week cigarette smoke (CS) exposed mice (WT $n = 10$; $Il36r^{-/-}$ $n = 10$). **b–d** Neutrophil and macrophage counts, CXCL1, IL-1α, IL-1β, and GM-CSF protein concentrations in BALF from untreated ($n = 6$) and IL-36γ-exposed mice (intratracheal instillation) ($n = 7$). **e, f** Cxcl1, Il1a, Il1b, and Il36g mRNA expression in either **e** naive mouse alveolar macrophages (pooled $n = 15$ mice) and stimulated in vitro with no cytokines (−), IL-36γ, or IL-1α/IL-1β or **f** in mouse bone marrow-derived neutrophils (from $n = 4$ mice) in vitro stimulated with no cytokines (−), IL-36γ, GM-CSF, or IL-36γ+GM-CSF. **g** Il36r and Il1r mRNA expression in mouse bone marrow-derived neutrophils (from $n = 4$ mice) in vitro stimulated with no cytokines (−), IL-36γ, GM-CSF, and IL-36γ+GM-CSF. **a, e, f** *$P \le 0.05$, **$P \le 0.01$, ***$P \le 0.001$, ****$P \le 0.0001$ vs all other groups by one-way ANOVA and Tukey's correction. **b, c, d, g** **$P \le 0.01$, ***$P \le 0.001$, ****$P \le 0.0001$ vs untreated by $t$ test. **e** AMs were pooled from 15 mice, data are presented as (mean ± SEM) of technical triplicates.

limit (Suppl. Fig. 1d). Thus, GM-CSF might skew the responsiveness of neutrophils toward IL-36 relative to IL-1 through upregulation of the IL-36R.

We also determined whether IL-36γ alone or in combination with GM-CSF would increase the transcription of *CXCL1*, *IL1A*, *IL1B*, and *IL36G* in human neutrophils. In contrast to mouse neutrophils, GM-CSF and not IL-36γ induced mRNA expression for *CXCL1* (2-fold), *IL1A* (69-fold), *IL1B* (9-fold), and *IL-36G* (4-fold) relative to unstimulated neutrophils (Fig. 2a). Combining IL-36γ stimulation with GM-CSF did not result in any further increase in the mRNA expression for these cytokines (Fig. 2a). Furthermore, IL-36 did not increase *IL36R* gene expression in human neutrophils relative to untreated cells (Fig. 2b). However, exposing human neutrophils to GM-CSF resulted in a significant increase of *IL36R* (encoded by *IL1RL2* herein designated as *Il36r*) mRNA (Fig. 2b). *IL1R* mRNA was not induced after any of these stimulations (Fig. 2b). Finally, we exposed human lung epithelial cells to IL-36γ and found significantly increased protein secretion and mRNA expression of *CXCL1*, *IL36G*, *GM-CSF*, *IL1A*, and *IL1B* (Fig. 2c, d).

Together, these findings indicated that in the alveolar compartment IL-36γ can act as an upstream inflammatory driver to promote neutrophil recruitment and production of pro-inflammatory IL-1 family cytokines in neutrophils and macrophages either alone or in combination with GM-CSF.

**IL-36 acts as an upstream inflammatory driver of macrophages and fibroblasts.** We conditioned bone marrow-derived macrophages (BMDMs) from naive mice toward a putative lung interstitial phenotype by exposing them to transforming growth factor (TGF)-β[36,37] and GM-CSF[34,35,38] in combination with IL-36 cytokines or IL-1α or IL-1β for 24 h. GM-CSF induced a 55-fold increase in mRNA for *Il36r* in BMDMs (Fig. 3a) and CXCL1 protein release after IL-36 cytokine stimulation (77-fold) was about 10-fold higher in GMCSF/TGF-β conditioned macrophages relative to that observed in unconditioned BMDMs (Fig. 3b). We therefore continued using GM-CSF/TGF-β conditioned BMDMs. We detected a 140-fold increase in *Il36g* mRNA expression, a 10-fold increase in *Cxcl1* mRNA expression, 4495-fold increase for *Il1a* mRNA expression, and a 235-fold increase for *Il1b* mRNA expression in TGF-β/GM-CSF conditioned BMDMs stimulated with IL-36 relative to unstimulated BMDMs (Fig. 3c). In contrast, BMDMs not conditioned in GM-CSF/TGF-β had a reduced responsiveness to IL-36 cytokines with *Il36g* (6-fold), *Cxcl1* (1.4-fold), *Il1a* (1.7-fold), and *Il1b* (3.4-fold) only slightly increased after IL-36 cytokine stimulation (Suppl. Fig. 2a). As we observed

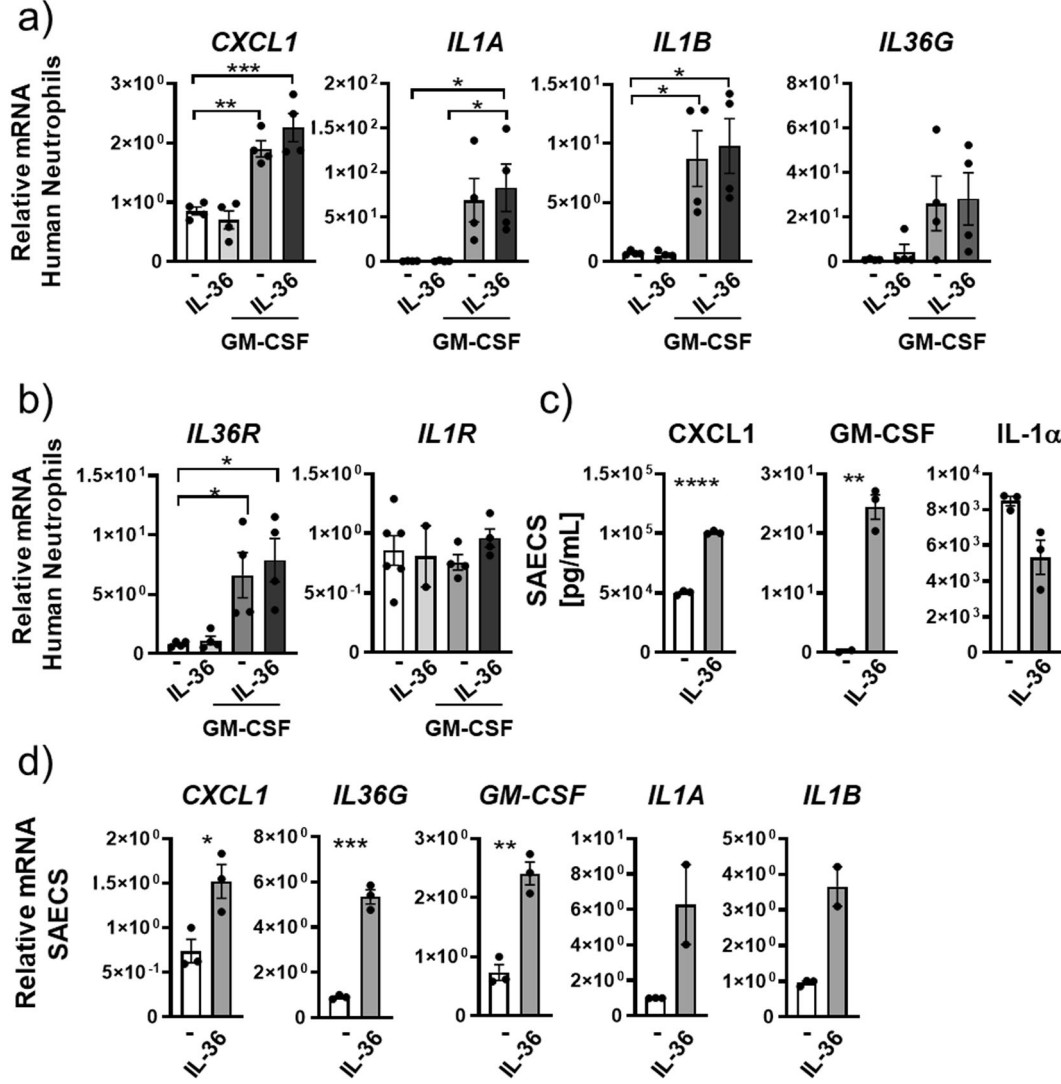

**Fig. 2 IL-36γ is an upstream inflammatory driver in human neutrophils and small airway epithelial cells. a, b** *CXCL1, IL1A, IL1B,* and *IL36G* mRNA expression in human peripheral blood-derived neutrophils (from n = 4 donors) in vitro stimulated with no cytokines (−), IL-36γ, GM-CSF, or IL-36γ+GM-CSF. **b** *IL36R* and *IL1R* mRNA expression in human peripheral blood-derived neutrophils (from n = 4 donors) neutrophils in vitro stimulated with no cytokines (−), IL-36γ, GM-CSF, and IL-36γ+GM-CSF. **c, d** CXCL1, GM-CSF, and IL1α protein concentrations and *CXCL1, IL36G, GM-CSF, IL1A,* and *IL1B* mRNA expression in small airway epithelial cells (SAEC) stimulated with either (−) or IL-36αβγ. **a, b** *P ≤ 0.05, **P ≤ 0.01, ***P ≤ 0.001, vs all other groups by one-way ANOVA and Tukey's correction. **c, d** *P ≤ 0.05, **P ≤ 0.01, ***P ≤ 0.001, ****P ≤ 0.0001 vs untreated by t test.

in AMs and neutrophils in Fig. 1, stimulation with a combination of IL-1α and IL-1β did not affect mRNA expression for *Cxcl1, Il1b,* and *Il1a* relative to untreated BMDMs (Fig. 3c and Suppl. Fig. 2a). Additionally, mouse BMDMs upregulated mRNA expression of *Il1r1* and *Il1rn* in response to IL-36γ stimulation, while *Il36rn mRNA* expression was undetected (Fig. 3c and Suppl. Fig. 2a).

In human monocyte-derived macrophages (MDMs) from healthy volunteer donors, *IL36R* mRNA was not induced by IL-36 cytokines or by GM-CSF stimulation (Fig. 4a). Stimulation with IL-36 in combination with GM-CSF significantly increased relative mRNA amounts of *IL1A* (53-fold) and *IL1B* (30-fold) (Fig. 4a). Stimulation of human MDMs with IL-36γ also promoted mRNA expression for *CXCL1* (Suppl. Fig. 3a). However, MDMs did not exhibit a significant increase in *IL36G* mRNA expression after IL-36 stimulation relative to untreated MDMs, which was only slightly increased (1.3-fold) after stimulation with GM-CSF. The combination of IL-36 cytokines and GM-CSF significantly increased the *IL36G* expression relative to GM-CSF alone (Fig. 4a).

When primary mouse fibroblasts obtained from the lungs of naive mice were exposed in vitro to IL-36γ or IL-1α and IL-1β stimulation for 24 h, we observed significantly increased mRNA expression for *Il1a* (284-fold), *Il1b* (337-fold), *Cxcl1* (151-fold), *Il36g* (50-fold), *Il1r1* (1.3-fold), *Il1Rn* (5-fold), and *Il36a* (22-fold) in response to IL-36γ relative to unexposed cells (Suppl. Fig. 2b). Stimulation with IL-1α and IL-1β did not result in any detectable increases in mRNA expression for these genes and *Il36b* and *Il36rn* were also not expressed in response to any of the stimulation conditions (Suppl. Fig. 2b). Conditioning fibroblasts with GM-CSF and TGF-β resulted in an additional increase in mRNA for *Il1a* (1015-fold), *Il1b* (761-fold), *Il36g* (81-fold), and *Il1rn* (6.2-fold) when exposed to IL-36 cytokines (Fig. 3d). Analogous to the findings in mouse neutrophils and mouse BMDMs, GM-CSF also increased mRNA expression for the *Il36r* in primary lung fibroblasts, albeit with a lower fold induction. IL-36 also slightly increased the *Il36r* in primary lung fibroblasts (Fig. 3e).

When primary human lung fibroblasts (from otherwise healthy donors) were stimulated with IL-36 cytokines, GM-

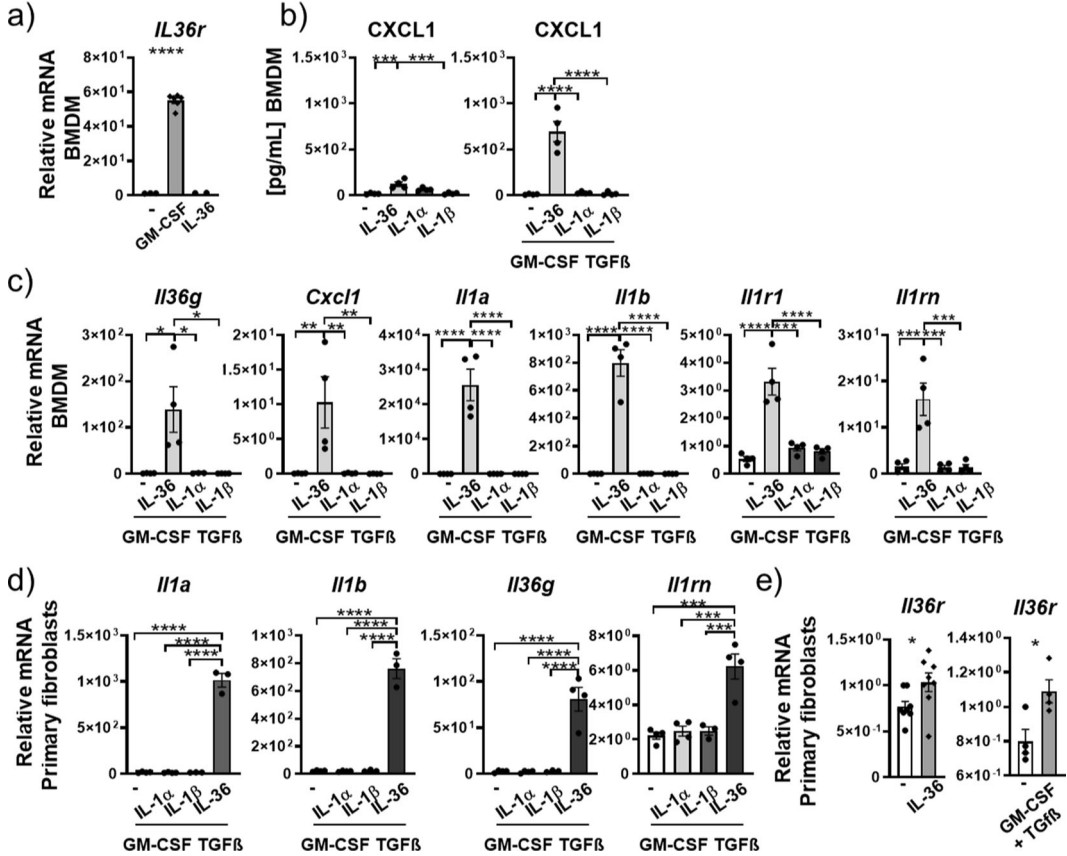

**Fig. 3 IL-36γ is an upstream amplifier in mouse macrophages and fibroblasts. a** Relative mRNA amounts of *Il36r* in naive mouse bone marrow-derived macrophages (BMDMs, $n = 6$) in response to no stimulation (−) or stimulation with GM-CSF and IL-36γ. **b** CXCL1 protein concentrations in supernatant of BMDMs ($n = 4$) after no stimulation (−), or stimulation with IL-36αβγ. IL-1α, IL-1β alone, or in combination with GM-CSF and TGFβ **c, d** Relative mRNA amounts of *Il36g, Cxcl1, Il1a, Il1b, Il1r1, Il1rn*, and *Il36rn* in BMDMs ($n = 4$) (**c**) and primary mouse fibroblasts ($n = 4$) (**d**) after no stimulation (−) or stimulation with IL-36αβγ, IL-1α, IL-1β alone, or in combination with GM-CSF and TGF-β. **e** Relative mRNA amounts in primary mouse fibroblasts ($n = 4$) of *Il36r* after no stimulation (−) or stimulation with IL-36αβγ or GM-CSF and TGF-β. Shown are the mean values ± SEM of biological replicates. **a–d** *$P \leq 0.05$, **$P \leq 0.01$, ***$P \leq 0.001$, ****$P \leq 0.0001$ vs all other groups by one-way ANOVA and Tukey's correction. **e** *$P \leq 0.05$, vs untreated by $t$ test.

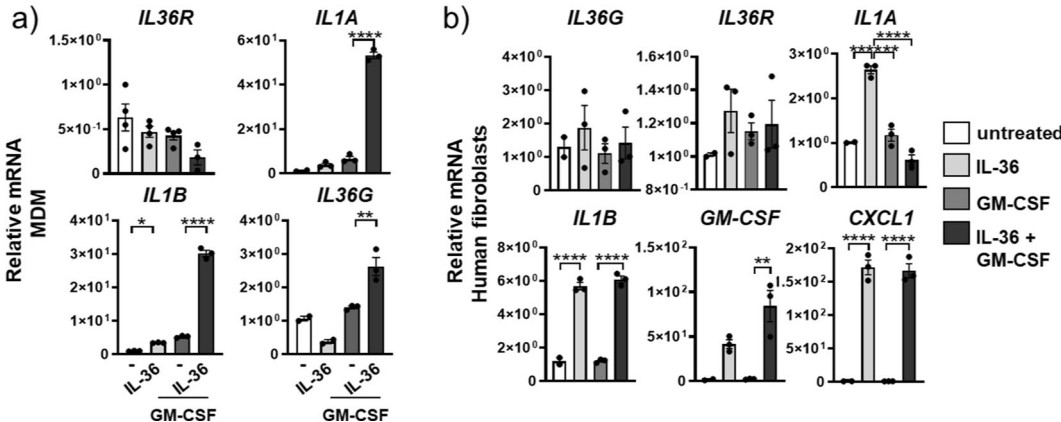

**Fig. 4 IL-36γ is an upstream amplifier in human macrophages and fibroblasts. a** Relative mRNA amounts in human monocyte-derived macrophages (MDM) (depicted are mean values ± SME of technical triplicates from one representative of four experiments) of *IL-36G, IL-36R, IL1A*, and *IL1B* and **b** IL-36γ, *IL-36R, IL1A, IL1B, GMCSF*, and *CXCL1* in primary human lung fibroblasts ($n = 4$) after no stimulation (−) or stimulation with IL-36αβγ, GM-CSF alone, or with the combination of IL-36αβγ and GM-CSF. Shown are the mean values ± SEM of biological replicates. **a, b** *$P \leq 0.05$, **$P \leq 0.01$, ***$P \leq 0.001$, ****$P \leq 0.0001$ vs all other groups by one-way ANOVA and Tukey's correction.

CSF, or the combination thereof, we observed increased mRNA expression for *IL36G, IL36R, IL1A, IL1B, GM-CSF*, and *CXCL1* after IL-36 stimulation relative to untreated fibroblasts (Fig. 4b).

Inflammatory conditions in the lung also promote tissue remodeling[39], in which matrix metalloproteinase 9 (MMP9) has been shown to be an important mediator[40]. MMP9 protein and mRNA expression were increased in both primary lung

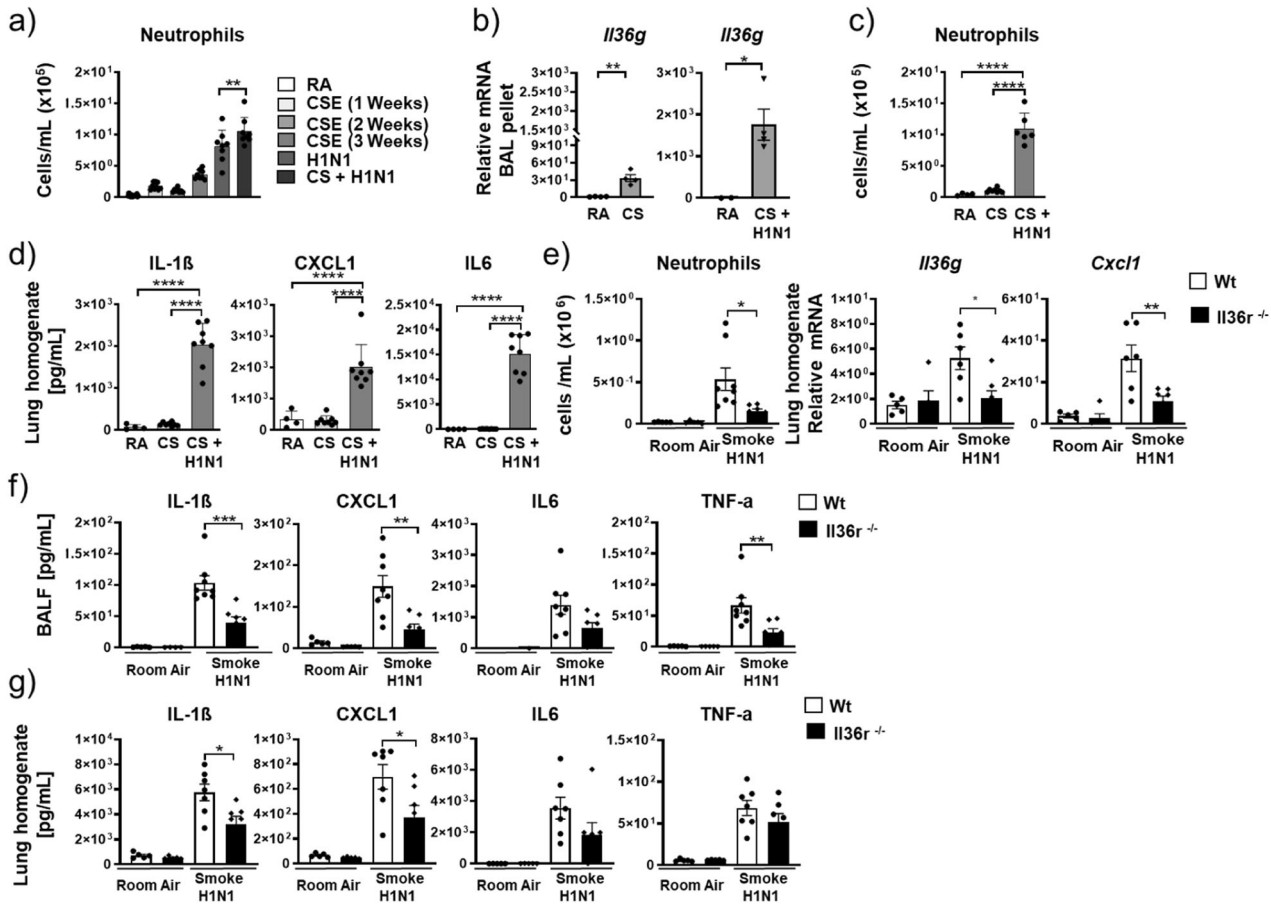

**Fig. 5 IL-36γ is critical in neutrophilic lung inflammation. a** Neutrophil counts in BALF from room air (RA) ($n = 18$) exposed or from 1-week ($n = 9$), 2-week ($n = 9$), and 3-week ($n = 8$) cigarette smoke (CS) exposed mice and from H1N1(4 days post treatment) exposed mice ($n = 8$) and from mice exposed to 2 weeks CS followed by 48 h of H1N1 exposure ($n = 8$). **b** Relative mRNA amounts of *IL-36g* in the cellular BAL pellet from room air (RA) and 2 week cigarette smoke (CS)-exposed mice ($n-4$). **c, d** Neutrophil numbers in BALF and IL1β, CXCL1, and IL-6 protein concentrations in lung homogenate of room air-exposed mice (RA; $n = 4$), 2-week CS-exposed mice ($n = 4$–8), and 2-week CS-exposed mice challenged with H1N1 for 48 h (CS + H1N1) ($n = 4$–8). **e** Neutrophil numbers in BALF and relative mRNA amounts of *IL-36g and Cxcl1* in lung homogenate from room air-exposed (RA, $n = 5$) and 2-week CS-exposed mice challenged with H1N1 for 48 h ($n = 7$–8) exposed WT and *Il36r*$^{-/-}$ mice. **f, g** IL1β, CXCL1, IL6, and TNF protein concentrations in BALF and in lung homogenate from room air-exposed (RA, $n = 5$) and 2-week CS-exposed mice challenged with H1N1 for 48 h ($n = 7$–8) exposed WT and *Il36r*$^{-/-}$ mice. Depicted are mean values ± SEM of biological replicates. **a**, **c–g** *$P \leq 0.05$, **$P \leq 0.01$, ***$P \leq 0.001$, ****$P \leq 0.0001$ vs all other groups by one-way ANOVA and Tukey's correction. **b** *$P \leq 0.05$, **$P \leq 0.01$, ***$P \leq 0.001$, ****$P \leq 0.0001$ vs untreated by $t$ test.

fibroblasts and BMDMs after stimulation with IL-36 cytokines (Suppl. Fig. 2c).

In summary, these findings highlight the difference between IL-1α/β and IL-36 activation and pinpoint IL-36 as an upstream amplifier of human and mouse macrophages and fibroblasts.

**IL-36 is important in neutrophilic lung inflammation in vivo.** Thus far, the data suggested that IL-36 signaling was important for neutrophil recruitment and activation both in vivo (Fig. 1a–c) and in vitro (Figs. 1 and 3). We therefore predicted that IL-36 would play an important role in a lung inflammatory condition associated with high neutrophil numbers. To test this hypothesis, we advanced our CS smoke model depicted in Fig. 1 to either CS for 1, 2, or 3 weeks or to CS (for 2 weeks) followed by exposure to H1N1 influenza virus for 48 h (CS + H1N1). Neutrophil numbers in recovered BAL were $1.8 \times 10^5$/mL after 1 week CS exposure, $8 \times 10^4$/mL after 2 weeks of CS exposure, $3.6 \times 10^5$/mL after 3 weeks of CS exposure, $8.2 \times 10^5$/mL after H1N1, and $1.05 \times 10^6$/mL after CS + H1N1 (Fig. 5a, c). Importantly, consistent with the higher numbers of neutrophils in CS + H1N1 mice relative to CS

mice, CS + H1N1-exposed mice exhibited significantly higher increases in *Il36g* mRNA in the BAL pellet (1759-fold increased relative to RA) compared to CS-exposed mice (40-fold relative to RA) (Fig. 5b). In addition, as indicators of lung inflammation, we found a significant increase of IL-1β, CXCL1, and IL-6 protein in lung tissue of CS + H1N1-exposed mice relative to CS-exposed mice (Fig. 5d). These findings showed that this model was suitable to generate lung inflammatory conditions with high neutrophil recruitment to the alveolar space, which also correlated with marked *Il36g* expression.

We therefore utilized this model to determine the contribution of IL-36 cytokine signaling to lung inflammation in vivo. *Il36r*$^{-/-}$ mice exposed to CS + H1N1 exhibited a significant decrease (~72%) of BAL neutrophil numbers relative to that observed in CS + H1N1-exposed WT mice (Fig. 5e). *Il36r*$^{-/-}$ mice exposed to CS + H1N1 also exhibited significantly decreased mRNA expression for *Cxcl1* in the lung homogenate relative to CS + H1N1-exposed WT mice (Fig. 5e). Furthermore, *Il36r*$^{-/-}$ mice exposed to CS + H1N1 also exhibited significantly decreased mRNA expression for *Il36g* in the lung homogenate (Fig. 5e). Finally, IL-1β, CXCL1, IL6, and TNF-α protein concentrations were also

decreased in CS + H1N1-exposed $Il36r^{-/-}$ mice in both BAL fluid (Fig. 5f) and lung homogenate (Fig. 5g and Suppl. Fig. 4a) relative to CS + H1N1-exposed WT.

We next used mice with genetic deficiency in IL-1RAP (the common receptor chain for the receptors for IL-1α, IL1-β, IL-33, and IL-36[2]) to define the effect of IL-36R signaling (reflecting the contribution of IL-36) within the IL-1 family in this model. CS + H1N1-exposed $Il1rap^{-/-}$ exhibited a significant reduction in the expression of markers for lung inflammation (Suppl. Fig. 4b). Specifically, we observed a 61% reduction in BAL neutrophil numbers relative to WT mice exposed to CS + H1N1. In addition, IL-1β, CXCL1, and IL6 protein concentrations in lung homogenate were significantly reduced in CS + H1N1-exposed $Il1rap^{-/-}$ mice relative to CS + H1N1-exposed WT mice (Suppl. Fig. 4b). Importantly, the percentage of reduction in BAL neutrophil numbers was comparable between $Il1rap^{-/-}$ and $Il36r^{-/-}$ CS + H1N1-exposed mice (Suppl. Fig. 4c). While $Il1rap^{-/-}$ CS + H1N1-exposed mice exhibited an 81% reduction for CXCL1 and an 80% reduction for IL-1β protein concentrations in lung homogenate, $Il36r^{-/-}$ CS + H1N1-exposed mice exhibited a 47% reduction for CXCL1 and 44% reduction for IL-1β. Notably, the percentage of reduction in IL-6 protein amounts in the lung homogenate was equal between $Il1rap^{-/-}$ (45%) and $Il36r^{-/-}$ (48%) CS + H1N1-exposed mice (Suppl. Fig. 4c). These findings indicated that, within the IL-1 family of cytokines that signal through IL-1RAP, IL-36 cytokines contribute to a large degree to the lung inflammatory response in CS + H1N1 mice.

**IL-36 cooperates with polyinosinic-polycytidylic acid (poly(I:C)) on macrophages and fibroblasts.** Based on the observed attenuated inflammation in $Il36r^{-/-}$ mice in response to CS + H1N1, we next employed in vivo approaches in which we used the Toll-like receptor 3 (TLR3) agonist poly(I:C) as a viral mimetic and focused on measuring neutrophils, GM-CSF, CXCL1, IL-36, and MMP9 cytokine expression in the BAL of mice. Mice exposed to i.t. poly(I:C) had increased neutrophils (but no increases in macrophages) in the BAL, 12 and 24 h post poly(I:C) exposure relative to PBS-exposed mice (Suppl. Fig. 5a). In addition, we found significantly increased protein concentrations for CXCL1 (at 4 and 24 h), GM-CSF (at 24 h), and MMP9 (at 24 h) in BAL, as well as increased $Il36g$ mRNA in the BAL pellet (at 12 h) relative to PBS-exposed mice (Suppl. Fig. 5a, b).

We next exposed AMs obtained by BAL and bone marrow-derived neutrophils from naive mice to poly(I:C) in vitro. Poly(I:C) induced small but significant increases in mRNA expression for $Il36g$ (2-fold), $Cxcl1$ (2.3-fold), Il1a (1.16–fold), $Il1b$ (2.17-fold), and $Il36r$ (1.11-fold) in AMs relative to untreated controls (Suppl. Fig. 5c). In mouse bone marrow-derived neutrophils, poly(I:C) alone failed to induce increased mRNA expression for $Il36g$, $Cxcl1$ Il1a, and $Il1b$ (Fig. 6a). Combining poly(I:C) stimulation with IL-36γ stimulation resulted in numerical increased mRNA expression for $Il1b$ but had similar effects on $Cxcl1$ and trended toward increased expression of $Il1a$ (Fig. 6a).

Combining IL-36 cytokines with poly(I:C) resulted in significantly increased relative mRNA amounts in BMDMs and primary mouse fibroblast for $Il36g$, $Cxcl1$, $Il1a$, $Il1b$, and $Gmcsf$ (Fig. 6b) and increased protein concentrations for CXCL1, IL-36γ (pro-form), GM-CSF, and MMP9 relative to the individual stimulations compared to untreated cells (Suppl. Fig. 5d, f). Interestingly, as observed for AMs (Suppl. Fig. 5c), poly(I:C) promoted mRNA expression for the $Il36r$ on BMDMs (Suppl. Fig. 5e). Primary mouse lung fibroblasts were comparable to BMDMs (Fig. 6b). Of note, poly(I:C) alone failed to induce $Il36g$, $Cxcl1$, $Il1a$, $Il1b$, $Gmcsf$, and $Il36r$ mRNA in primary lung fibroblasts, highlighting the requirement for cooperating with IL-

36 in activating fibroblasts (Fig. 6b and Suppl. Fig. 5e, f). The combination of poly(I:C) and IL-36 (7.5-fold) resulted in a 50% reduced $Ifnb1$ mRNA expression in BMDMs relative to poly(I:C) (15-fold) stimulation alone. IL-36 alone did not induce $Infb1$ expression in BMDMs (Suppl. Fig. 5g). Neither poly(I:C) nor IL-36 stimulation resulted in increased $Infb1$ mRNA expression in fibroblasts (Suppl. Fig. 5g).

Similar to mouse fibroblasts, human lung fibroblasts stimulated with the combination of poly(I:C) and IL-36 cytokines exhibited a significant increase in $CXCL1$, $IL1A$, $IL1B$, and $GM-CSF$ mRNA relative to untreated controls (Fig. 7a). Neither poly(I:C) nor IL-36 cytokine exposure promoted $IL36G$ mRNA expression in human fibroblasts (Fig. 7a).

In human MDMs, IL-36 cytokine stimulation resulted in significant increase of $CXCLl1$, $IL1A$, and $IL1B$ mRNA expression relative to untreated or poly(I:C) alone and was not changed by combining IL-36 cytokines with poly(I:C) (Fig. 7b). Similar to the fibroblast data, $IL36G$ mRNA expression in MDMs was not increased after IL-36 cytokine, poly(I:C), or IL-36 cytokine + poly (I:C) stimulation (Fig. 7b).

Human basal lung epithelial cells exposed to poly(I:C) exhibited significant and marked increases in protein concentrations of GM-CSF (63-fold), CXCL1 (2.1-fold), and IL-36γ (71-fold) and mRNA expression of $GM-CSF$ (7-fold), $CXCL1$ (4-fold), $IL36G$ (364-fold), $IL1A$ (186-fold), and $IL1B$ (49-fold) (Fig. 8a, b). LPS, used as an additional TLR agonist did not induce mRNA increases of these cytokines. (Fig. 8a, b). IL-1α protein concentration was not increased after poly(I:C) or LPS stimulation relative to untreated control and IL-1β was only detected after poly(I:C) stimulation (Fig. 8a).

These data indicated that poyl(I:C) increased the production of pro-inflammatory cytokines in lung epithelial cells and cooperated with IL-36 to further amplify pro-inflammatory pathways.

**Neutrophils are a source of IL-36γ in acute lung injury.** To determine neutrophils as a source of IL-36 not only in a lung inflammation model generated by CS or CS and virus, we also examined a lung inflammation model simulating exposure to bacterial stimuli. We therefore exposed mice to aerosolized LPS for 4 h before proceeding with single-cell RNA sequencing (scRNA-seq) analysis of the BAL cells. We chose 4 h exposure as it was determined to be the earliest time point with maximal neutrophil numbers in the BAL fluid ($2.12 \times 10^6$ neutrophils/mL) in a pilot experiment (Suppl. Fig. 6a). After LPS challenge, the majority of the sequenced BAL cells represented neutrophils, which accounted for 77% of all BAL cells, while AMs represented only 18% of the BAL cell population (Fig. 9a). Additionally, we identified natural killer cells (4%) and few erythrocytes (1%) based on their reported characteristic gene expression profiles[41] (Fig. 9a and Suppl. Fig. 6b, c). We then determined the relative transcriptional expression of $Il36a$, $Il36b$, and $Il36g$ in comparison to $Il1a$ and $Il1b$ in BAL neutrophils and macrophages. High-resolution scRNA-seq analysis allowed us to identify neutrophils as the main source of $Il36g$. $Il36g$ was detected in 84% of the neutrophils and its transcript levels were 10.5-fold increased relative to those detected in AMs (Fig. 9b, d). In contrast, $Il1b$ was strongly expressed by both neutrophils and AMs. Nevertheless, $Il1b$ was still increased 3.6-fold in neutrophils relative to AM while $Il1a$ was detected in both populations, whereas the overall expression was significantly (1.5-fold) increased in neutrophils relative to AMs (Fig. 9b). In contrast to $Il36g$, $Il36a$ was only detected in very few single cells (while $Il36b$ was not detected in any cell type (Suppl. Fig. 6d)). Next, we investigated the expression of $Cxcl1$ across the different cell types. $Cxcl1$ expression was predominantly restricted to AMs and increased 1.9-fold

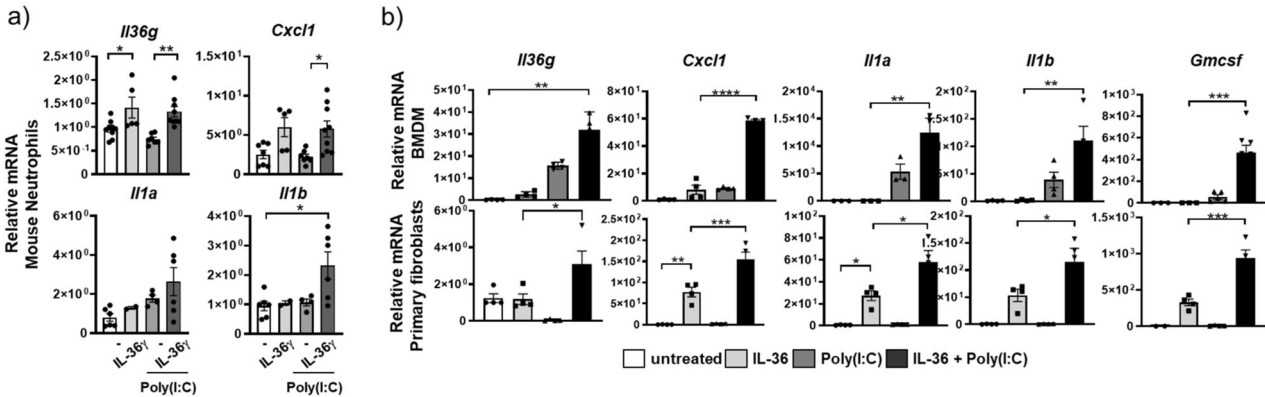

**Fig. 6 IL-36γ cooperates with poly(I:C) on mouse macrophages and fibroblasts. a** Relative mRNA amounts of *Il36g*, *Cxcl1*, *Il1a*, and *Il1b* in mouse neutrophils ($n = 2$–6) unstimulated with (−) or stimulated with IL-36αβγ, poly(I:C), or the combination of IL-36αβγ and poly(I:C) (depicted are mean values ± SEM of biological replicates). **b** Relative mRNA amounts of *Il36g*, *Cxcl1*, *Il1a*, *Il1b*, and *Gmcsf* in BMDMs ($n = 4$) and primary mouse fibroblasts ($n = 4$) (depicted are mean values ± SEM of biological replicates) unstimulated (−) or stimulated with IL-36αβγ, poly(I:C), or the combination of IL-36αβγ and poly(I:C). *$P \leq 0.05$, **$P \leq 0.01$, ***$P \leq 0.001$, ****$P \leq 0.0001$ vs all other groups by one-way ANOVA and Tukey's correction.

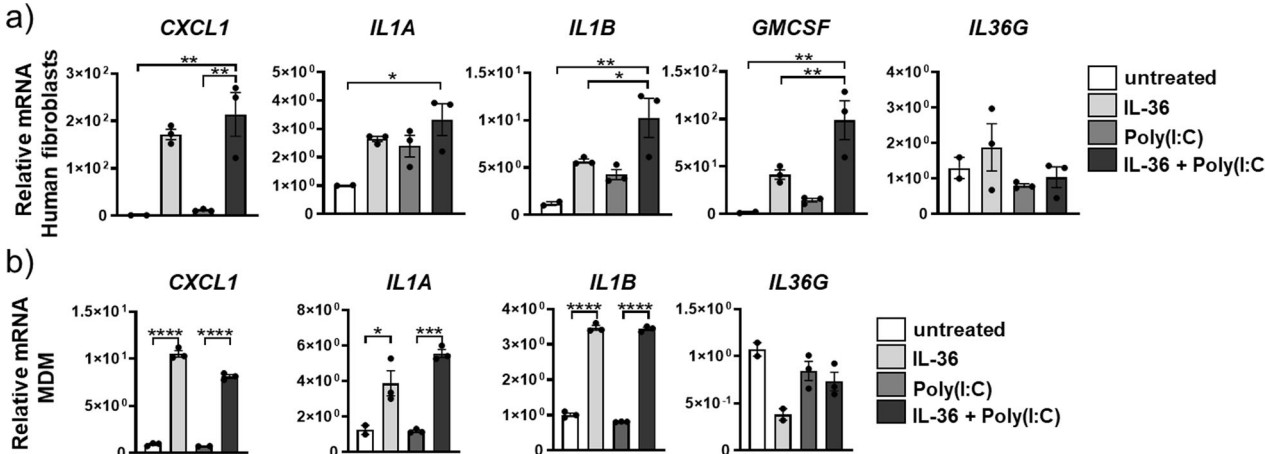

**Fig. 7 IL-36γ cooperates with poly(I:C) on human macrophages and fibroblasts. a, b** Relative mRNA amounts of *IL-36g*, *CXCLl1*, *IL1A*, *IL1B*, and *GM-CSF* in human fibroblasts and human MDMs (depicted are mean values ± SME of technical triplicates from one representative of three to four experiments) unstimulated (−) or stimulated with IL-36αβγ, poly(I:C), or the combination of IL-36αβγ, and poly(I:C). *$P \leq 0.05$, **$P \leq 0.01$, ***$P \leq 0.001$, ****$P \leq 0.0001$ vs all other groups by one-way ANOVA and Tukey's correction.

compared to neutrophils (Fig. 9b). We next determined mRNA expression for *Il36a,b,g* and *Il1a,b* as well as *Cxcl1* by quantitative polymerase chain reaction (qPCR) in mouse bone marrow-derived neutrophils and mouse AMs from naive mice in response to in vitro stimulation with LPS for 4 h to closely reflect the in vivo stimulation protocol. Consistent with the scRNA-seq data, we found that neutrophils were more responsive to LPS stimulation in terms of *Il36g* expression (48-fold increased) compared to the expression in AMs (8-fold increased) (Fig. 9c). In addition, induction of *Cxcl1* mRNA expression upon LPS stimulation was restricted to AM and not induced in stimulated neutrophils (Fig. 9c). *Il1a* and *Il1b* were induced in both cell types after LPS stimulation but more predominantly in AMs. AMs had increased *Il36a* (690-fold) mRNA expression after LPS stimulation and *Il36b* was undetectable (Suppl. Fig. 6e). Furthermore, IL-36α and IL-36β mRNA expression remained below the detection limit for neutrophils.

Human neutrophils and human MDMs exhibited a 100- and 200-fold increase in *IL36G* mRNA expression after LPS stimulation relative to unstimulated cells. Human fibroblasts exhibited a 2.5-fold *IL36G* mRNA induction after LPS stimulation relative to untreated controls (Fig. 9e).

A more detailed analysis of the neutrophil clusters in the scRNA-seq analysis revealed two subpopulations that we designated N1 and N2 (Suppl. Fig. 6f). To differentiate the subclusters, we did a pathway analysis. Subcluster N2 showed increased IL-1 family and IFN pathways, whereas subcluster N1 exhibited increased remodeling pathways (Suppl. Fig. 6g). Finally, increased differential expression of MMP9 was observed to be restricted to subcluster N1, suggesting that these neutrophils could promote tissue remodeling (Suppl. Fig. 6h).

A hypothetical model of how cells and mediators interplay with IL-36 to promote neutrophilic lung inflammation is depicted in Fig. 10.

## Discussion

Here, we combined chronic and acute lung inflammation mouse models induced by CS, viral (poly(I:C)), and bacterial (LPS) components with additional in vivo and in vitro approaches using primary mouse and human cells to provide mechanistic insight that identifies IL-36 within the IL-1 family of cytokines as an early upstream innate immune driver and amplifier of acute and chronic lung inflammation, particularly when associated with

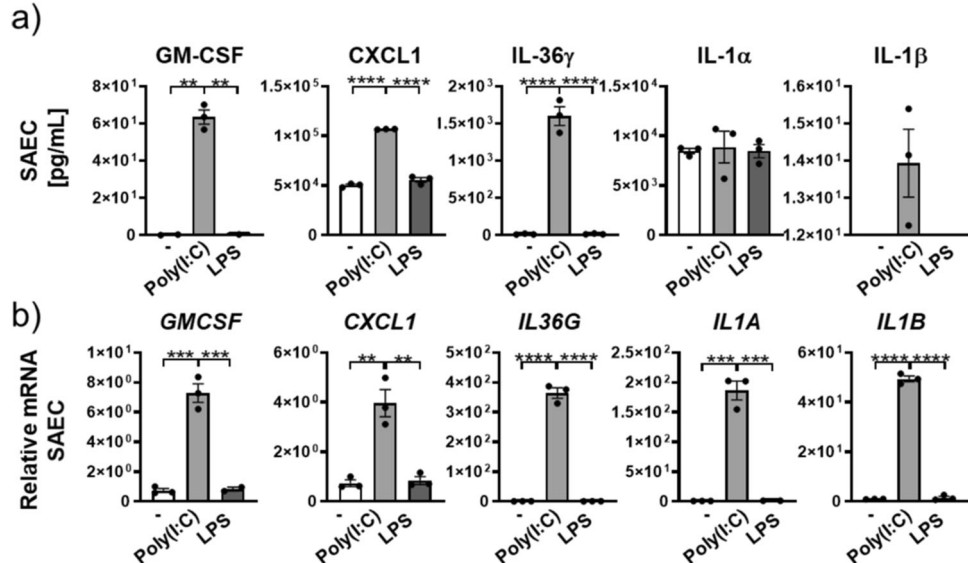

**Fig. 8 Small airway epithelial cells as the main cellular source of GMCSF. a** GM-CSF, CXCL1, IL-36γ, IL-1β, and IL-1α protein concentrations and *GM-CSF, CXCL1, IL36G, IL1A,* and *IL1B* mRNA expression of lung epithelial basal cells from healthy donor stimulated with poly(I:C) and LPS (depicted are mean values ± SEM of technical triplicates from one representative of three experiments). *$P \leq 0.05$, **$P \leq 0.01$, ***$P \leq 0.001$, ****$P \leq 0.0001$ vs all other groups by one-way ANOVA and Tukey's correction.

neutrophils. Mechanistically, lung neutrophils were identified as both a cellular source and target of IL-36 and IL-36 promoted the pro-inflammatory activation of lung macrophages, fibroblasts, and epithelial cells. IL-36 stimulated feed-forward signaling that resulted in the generation of IL-36 and IL-1 in fibroblasts and macrophages. Moreover, IL-36 cooperated with viral analog poly (I:C) and with GM-CSF to enhance the pro-inflammatory responses of macrophages and fibroblasts.

There is substantial evidence associating the pro-inflammatory role of IL-36 with the presence of neutrophils in mouse models of skin inflammation and also in human skin diseases[8,42], such as GPP or hidradenitis suppurativa, where blockade of IL-36 signaling can greatly reduce disease symptoms[20]. In addition, IL-36 has been described to be increased in patients with COPD[43], and in COPD, neutrophils are correlated with the severity of the symptoms and with microbial exacerbation, while high neutrophil numbers are correlated with poor prognosis and disease progression[44,45]. Previous reports also showed GM-CSF release after pulmonary infection and the importance of GM-CSF in inflammatory processes in the lung has been reported, including in COPD[46,47]. Finally, previous reports had suggested an important role for IL-36 in mouse models of bacterial and viral pneumonia[7,12]. Yet, no studies had investigated the interplay between IL-36, GM-CSF, and neutrophils in chronic lung inflammation or in lung inflammation in which acute virus challenge is superimposed on chronic injury, a reductionist mechanistic model for human lung disease secondary to smoking and virus infection.

Using a genetic approach employing mice with *Il36r* deficiency, we showed that IL-36 signaling was a critical component of the pro-inflammatory response in both CS- and CS + HN1N-exposed mice, demonstrating a role for IL-36 in both low-grade chronic lung inflammation (CS model) as well as high-grade acute microbial lung inflammation super-imposed on pre-existing chronic injury, such as during acute viral exacerbations in COPD (CS + H1N1 model). Specifically, *Il36r*−/− mice had significantly attenuated lung neutrophil influx and reduced alveolar IL-1β, CXCL1, IL-6, and TNF-α concentrations. Making use of *IL1rap*−/− mice, we isolated the contribution of IL-36 cytokines within the IL-1 family of cytokines in this model and showed that

the phenotype of the *Il1rap*−/− mice was largely replicated in *Il36r*−/− mice. These data solidify IL-36 as a key upstream pro-inflammatory driver among the IL-1 family of cytokines that signal through IL-1RAP. Importantly, genetic ablation of IL-36 signaling was sufficient to attenuate recruitment of neutrophils to the same degree as seen in *Il1rap*−/− mice, while the protein expression of IL-1β and CXCL1 was more attenuated in the *Il1rap*−/− mice. These findings can be interpreted that blocking IL-36R allows for maintaining at least part of the IL-1 innate-mediated immune response necessary to maintain some level of immune defense, while it abrogates the acute inflammatory and tissue destructive effects of IL-36, which superimpose on the underlying inflammation and exacerbate immune pathology and substantially compromise organ function in the face of acute pathogen challenge. The findings that *Il36g* expression within the alveolar immune cell compartment was restricted to neutrophils in LPS-challenged mice and that stimulation of mouse AMs, mouse neutrophils, and human neutrophils with LPS in vitro increased IL-36γ production demonstrate that neutrophils can also be an important source of IL-36γ in humans and mice in neutrophilic inflammation associated with bacterial infection. Thus, we propose that IL-36R blockade in patients with chronic lung disease prone to microbial exacerbations will alleviate symptoms while preserving sufficient innate immune signaling mediated by IL-1α and IL-1β to conserve host defense.

We also demonstrate that IL-36γ promoted generation of IL-36γ itself and was upstream of both IL-1α and IL-1β in neutrophils, fibroblasts, and macrophages. IL-36γ was also capable of inducing CXCL1 in vivo and in a variety of cell types in vitro, including lung macrophages, lung fibroblasts, neutrophils, and lung epithelial cells. Notably, CXCL1, as well as IL-1α and IL-1β, expression in macrophages was restricted to IL-36γ stimulation and not observed with the other IL-1 family cytokines IL-1α and IL-1β, placing an important role on IL-36 in an upstream amplification loop for the production of CXCL1 and IL-1 family cytokines in a variety of innate immune cells. Neutrophils themselves also responded to IL-36 stimulation with CXLC1 production, highlighting a feed-forward loop between IL-36, CXCL1, and neutrophils. Thus, the IL-36γ-triggered pro-

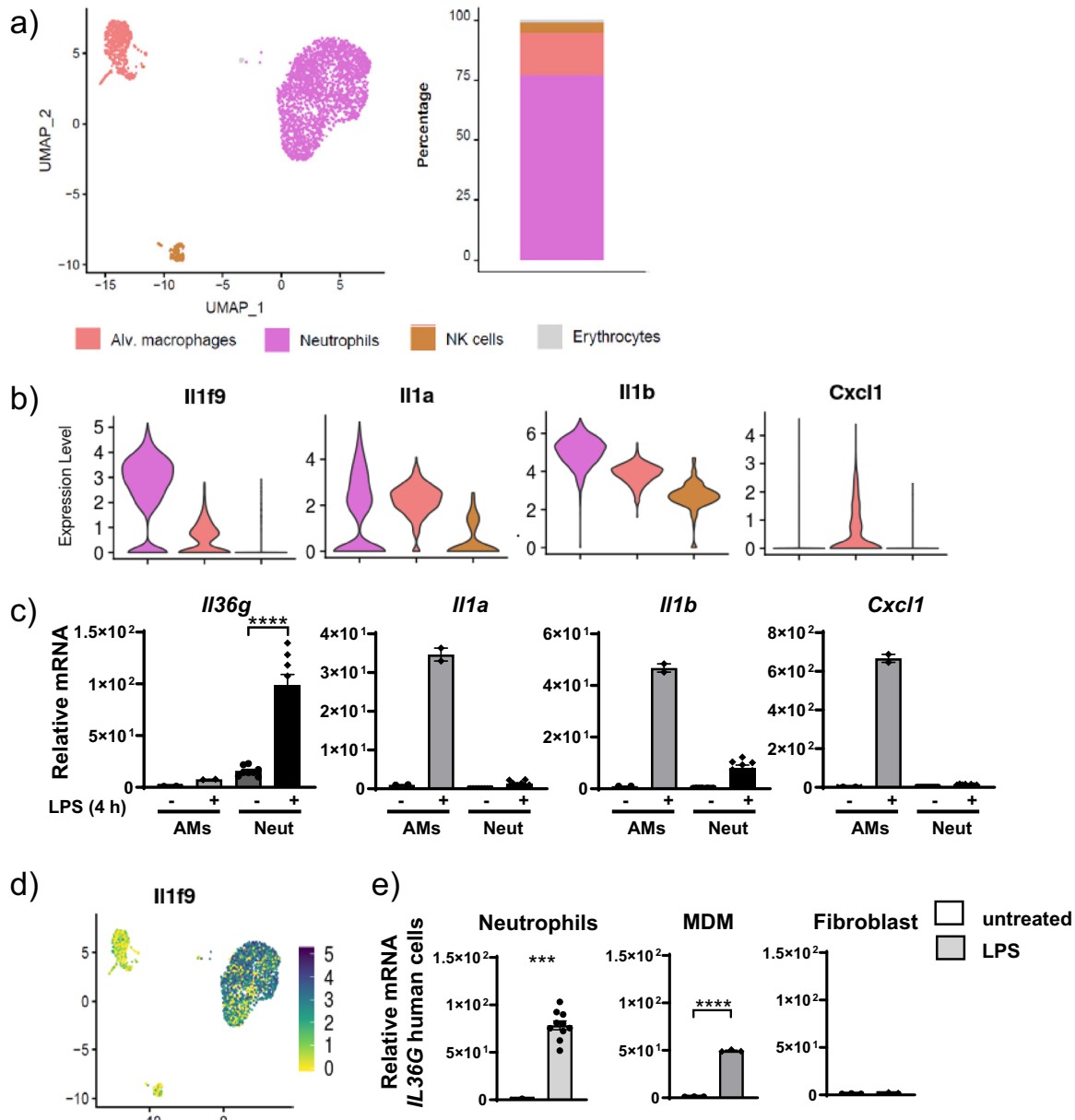

**Fig. 9 Neutrophils are a source of IL-36γ in acute lung injury. a** UMAP representation of the cell populations identified in BAL from LPS-exposed mice. Cell identity is indicated by the color code. Relative frequency of the cell types is shown in the bar plot using the same color code. **b** Violin plots display normalized expression levels of *Il36g* (*Il1f9*), *Il1a*, *Il1b*, and *Cxcl1* across the different cell populations. **c** Relative mRNA amounts of the same genes used in **b** after 4 h in vitro LPS stimulation of naive mouse alveolar macrophages (AM) and naive mouse bone marrow-derived neutrophils; depicted are mean values ± SEM of biological duplicates (each biological duplicate represents pooled AMs from 4 mice) and *n* = 8 from the neutrophils analyzed by one-way ANOVA. **d** Visualization of normalized *IL-36g* (*Il1f9*) expression levels per single cell. **e** mRNA expression of *IL-36g* after LPS stimulation in vitro of human neutrophils, MDMs (depicted are mean values ± SME of technical triplicates from one representative of three to four experiments) and human fibroblasts *n* = 4. Data represents the depicted mean ± SEM of biological replicates; ***P ≤ 0.001, ****P ≤ 0.0001 vs untreated by *t* test.

inflammatory mediator output broadly facilitates feed-forward amplification of the inflammatory response by engaging a variety of cells to increase neutrophil recruitment (CXCL1) and sensitization to IL-36γ or IL-1.

Our findings furthermore demonstrate that, downstream of neutrophils, activated IL-36 cytokines subsequently cooperate with additional pro-inflammatory signals, such as GM-CSF or poly(I:C)/viruses as an upstream driver of IL-1, GM-CSF, and CXCL1, further amplifying the pro-inflammatory interplay between neutrophils, IL-36, IL-1, and GM-CSF. Moreover, mice exposed to poly(I:C) had increased neutrophils and protein concentrations of GM-CSF and CXCL1 in the BAL. Notably, poly

(IC) did not induce GM-CSF or CXCL1 in macrophages or neutrophils, making the lung epithelium the most likely source of GM-CSF and CXCL1 in vivo in response to poly(I:C). Therefore, poly(I:C) can promote IL-36 generation indirectly through epithelial-derived cytokines such as GM-CSF, which acts on neutrophils to produce IL-36. Another important finding of our study was that IL-36 also cooperated with poly(I:C) to directly increase the expression of *Il36g*, *Cxcl1*, *Il1a*, *Il1b*, and *Gmcsf* in macrophages and fibroblasts. Thus poly(I:C) can also promote IL-36 signaling. Previous reports had suggested the cooperation between TLR signaling and cytokine signaling with IL-36 in promoting inflammation[32] (Fig. 6).

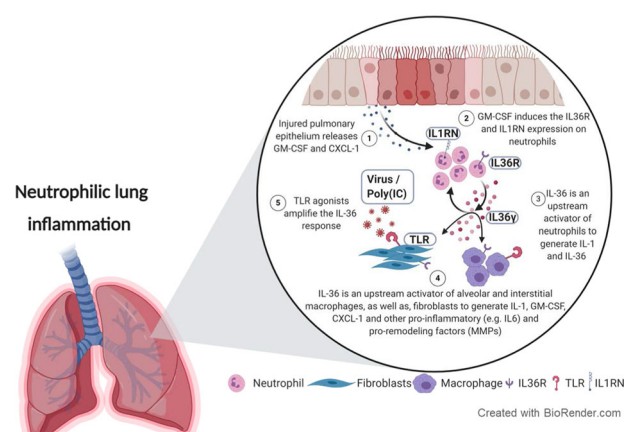

**Neutrophilic lung inflammation**

**Fig. 10 Proposed signaling pathways involving IL-36 as an upstream amplifier in neutrophilic lung inflammation. 1** Chronic (e.g., cigarette smoke) or acute (H1N1 or poly(I:C)) injury to the lung epithelium promotes release of CXCL1 and GMCSF. **2** CXCL1 promotes neutrophil recruitment while GM-CSF upregulates the IL-36R and IL1RN by neutrophils and promotes IL-36γ expression. **3** IL-36γ activates neutrophils to generate IL1 and IL36. **4** IL-36γ subsequently activates alveolar and interstitial macrophages and fibroblasts to generate IL-1, CXCL1, GM-CSF, MMPs, and more IL-36. GM-CSF also upregulates the IL36R on macrophages and on fibroblasts and sensitizes cells to IL-36. **5** Poly(I:C) as a TLR agonist cooperates with IL-36γ in amplifying activation of macrophages and fibroblasts. Note that IL-36RN is not induced while IL1RN is induced, shifting the balance toward heightened responsiveness to IL-36.

We also identified human lung epithelial cells as a source of IL-36γ in response to poly(I:C) but not LPS exposure (Fig. 8). There is literature evidence for IL-36γ expression in the lung epithelium in patients with lung disease[31,48]. Furthermore, IL-36γ is prominently expressed in keratinocytes of the skin[9,11], particularly in response to bacterial challenge or the presence of neutrophils[8,49]. Therefore, the lung epithelium might be an additional source of IL-36γ in the face of viral exacerbation, while the LPS-mediated IL-36 generation in neutrophils described here suggests neutrophils as a major source of IL-36 in the face of bacterial exacerbation.

In human neutrophils, *IL36G* mRNA was increased in response to IL-1 rather than IL-36γ. Therefore, it is tempting to speculate that in humans with underlying chronic lung inflammation the increased levels of IL-1[50] induce IL-36 signaling to promote tissue inflammation. The finding that IL-36R expression in murine neutrophils and human neutrophils was increased by GM-CSF and the observation that combining GM-CSF with IL-36 (mouse) or IL-1 (human) amplified the production of IL-36 suggests that GM-CSF tailors responsiveness of mouse and human cells toward IL-36 and IL-1, respectively. This suggests a tightly regulated and context-dependent restriction of IL-1 family cytokine generation and signaling. Context-dependent activity of IL-36 has been described previously by showing that the activity of IL-36 cytokines is ~1000-fold increased after extracellular processing by neutrophil-derived proteases, enabling IL-36 activity in the presence of neutrophils[16]. Furthermore, it was recently reported that neutrophil extracellular traps facilitate generation of IL-1 family members, particularly IL-36[51]. Thus neutrophils are important for both generating and activating IL-36, placing neutrophils as an early upstream pro-inflammatory event in the IL-36 signaling cascade.

Here, we made an additional observation that GM-CSF was an inducer of IL-36R in mouse and human neutrophils. Intriguingly, GM-CSF promoted the expression of IL-36R but not the IL-1 receptor on macrophages and fibroblasts. There is evidence for a

relationship between GM-CSF and neutrophils by the fact that humans treated with GM-CSF had increased numbers of neutrophils[52]. Another study demonstrated that LPS-exposed GM-CSF$^{-/-}$ mice had reduced infiltration of neutrophils compared to the WT control[53].

We also observed that IL-36γ- or IL-1-stimulated cells increased the expression of the IL-1RN but not the IL-36RN, suggesting unrestrained IL-36γ signaling while limiting IL-1 signaling in these conditions. Intriguingly, severe neutrophilic skin diseases in humans are associated with polymorphisms in IL-36RN genes[54,55].

Finally, inflammatory conditions in the lung are associated with tissue remodeling. Therefore, we used fibroblasts as structural target cells reported to produce MMP9 and examined MMP9 production as a surrogate for an important remodeling factor after IL-36 cytokine stimulation[56,57]. Consistent with previous reports, IL-36 was indeed capable of inducing MMP9 expression in primary lung fibroblasts[57], involving IL-36 signaling in pro-fibrotic fibroblast activity[21,56].

In conclusion, we have used several mouse models to reflect important triggers of lung inflammation that bear physiological relevance in humans with COPD, such as CS, CS + H1N1, poly(I:C), and LPS, allowing us to mechanistically investigate the interplay between IL-36 and neutrophilic lung inflammation. We have shown that injured epithelium released GM-CSF, which increased IL-36 expression in the alveolar compartment. IL-36 subsequently activated alveolar and interstitial macrophages, as well as fibroblasts, to generate IL-1, CXCL1, GM-CSF, MMPs, and IL-36 (Fig. 10). Our study thus pinpoints IL-36 as a key upstream pro-inflammatory driver and amplifier of neutrophilic lung inflammation and provides a mechanistic explanation for the link between IL-36 and neutrophilic inflammation that had been observed previously in human diseases and in animal models[19,29,57–59] (Fig. 10). Our data therefore provide an experimental rationale for exploring the therapeutic potential of IL-36 signaling blockade to attenuate pro-inflammatory events in human lung disease with a significant contribution of neutrophils, such as asthma and COPD.

## Methods

**Animals**. Female and male WT C57BL/6J mice and *Il1rap*$^{-/-}$ mice on the C57BL/6 background or *Il36r*$^{-/-}$ mice or WT Balb/cAnCrl mice, all 8–13 weeks of age, were purchased from Charles River (Sulzfeld, Germany or the US). Animals were housed in groups of 5 mice per cage under specific pathogen-free conditions in isolated ventilated cages at 20–25 °C and a humidity of 46–65% with a dark/night cycle of 12 h. Mice had free access to water and chow. All experiments were approved by the animal welfare officers within Boehringer Ingelheim Pharma GmbH and Co KG, as well as by the local authorities for the care and use of experimental animals (Regierungspräsidium Tübingen; TVV 12-009-G and 14-016-G; 35/9185.81-8). Experiments with influenza virus were performed under biosafety level 2 conditions and were in accordance with German national guidelines and legal regulations.

**Reagents**. Human and mouse primary cells in this study were stimulated with the cytokines and the concentrations depicted in Table 1. Unless otherwise described, the cells were stimulated with macrophage media containing Dulbecco's modified Eagle's medium (DMEM) (1×)+GlutaMAXTM-I (GIBCO #31966-021); 10% Spezial-HI fetal calf serum (FCS; GIBCO #16140-071); 1%NEAA (100x GIBCO #11140-035); 1% P/S (10,000 U/mL Penicillin, 10,000 μg/mL Streptomycin GIBCO #15140-122) and recombinant human MCSF.

**Poly(I:C) administration**. For i.t. administration of poly(I:C), female C57BL/6 mice, 8–10 weeks old (Javier Laboratories), were anesthetized with isoflurane for 3 min before instilling 2 mg/kg poly(I:C) (LMW) (#tlrl-picw/tlrl-picw-250, Invivo-Gen, Lot: PIW-40-03) diluted in NaCl (0.9%)) in a total volume of 50 μL/animal with a 1-mL syringe during inspiration. BAL was performed 4, 12, 24, and 48 h after the poly(I:C) administration.

**IL-36 i.t. administration**. For i.t. administration of IL-36γ, female Balb/c mice were anesthetized with isoflurane for 3 min and 1 μg IL-36γ (6996-IL/CF, Lot

**Table 1 Cytokines used in this study.**

| Species | Cytokine | Concentration | Catalog number | Manufacturer |
|---|---|---|---|---|
| Human/mouse | rhMCSF | 100 ng/mL (macrophage differentiation) 10 ng/mL (macrophage stimulation) | 216-MCC/CF | R&D Systems |
| Human/mouse | Salmonella LPS | 100 ng/mL | L6143-1MG | Sigma |
| Mouse | rmIL-36α | 33 ng/mL | 7059-ML/CF | R&D Systems |
| Mouse | rmIL-36β | 33 ng/mL | 7060-ML/CF | R&D Systems |
| Mouse | rmIL-36γ | 33 ng/mL | 6996-IL/CF | R&D Systems |
| Mouse | rmIL-1α | 10 ng/mL | 400-ML/CF | R&D Systems |
| Mouse | rmIL-1α | 50 ng/mL (Suppl. Fig. 1c) | 400-ML/CF | R&D Systems |
| Mouse | rmIL-1β | 10 ng/mL | 401-ML/CF | R&D Systems |
| Mouse | rmIL-1β | 50 ng/mL (Suppl. Fig. 1c) | 401-ML/CF | R&D Systems |
| Mouse | rmTNFα | 10 ng/mL | 410-MT/CF | R&D Systems |
| Mouse | rmGM-CSF | 10 ng/mL | 415-ML/CF | R&D Systems |
| Human/mouse | Poly(I:C) (LMW) | 1 µg/mL | tlrl-picw | InvivoGen |
| Human | rhTGFβ | 10 ng/mL | 7666-M/CF | R&D Systems |
| Human | rhIL-36α | 33 ng/mL | 6995-IL-010/CF | R&D Systems |
| Human | rhIL-36β | 33 ng/mL | 6834-ILB-025/CF | R&D Systems |
| Human | rhIL-36γ | 33 ng/mL | 2320-IL-025/CF | R&D Systems |
| Human | rhIL-1α | 10 ng/mL | 200-LA-010/CF | R&D Systems |
| Human | rhIL-1β | 10 ng/mL | 201-LB-010/CF | R&D Systems |
| Human | rhTNFα | 10 ng/mL | 410-MT/CF | R&D Systems |
| Human | rhGMCSF | 10 ng/mL | 7954-GM-010/CF | R&D Systems |

DAQQ041703A, R&D) that was diluted in PBS in a total volume of 50 µL/animal was administered with a 1-mL syringe during inspiration. BAL was performed 4 h after the IL-36γ administration. For time course analysis, mice were sacrificed 10, 20, or 30 min after the IL-36gγ administration (BAL was pooled from 2 to 3 animals from each group).

**LPS aerosol exposure.** Male Balb/c mice were exposed once to a single dose of aerosolized *Escherichia coli* LPS (Serotyp 055:B5, Sigma Aldrich) in PBS (1 mg/mL) for 30 min. LPS was administered by a nebulizer (Parimaster®) connected to a self-made Plexiglas box. After pre-flooding the box and all tubes for 30 min with the LPS aerosol, mice were transferred into the box and were exposed to a continuous flow of LPS aerosol for 25 min and left to remain for another 5 min after the aerosol was discontinued[60–62]. Mice were sacrificed 2, 4, 6, 8, 12, 18, 24, 36, 48, and 60 h after LPS exposure and BAL was performed as described in the section "Bronchoalveolar lavage."

**CS exposure model.** Female and male C57BL/6J mice were either exposed to RA or CS in a heated (38 °C) perspex box (homemade, whole-body exposure chamber) for 4 days (4–5 cigarettes/day), with the experimental readout on day 5 (1-week CS exposure model) or for 5 days/week (4–5 cigarettes/day) with the experimental readout on day 11 (2-week CS exposure model)[63]. For the 3-week CS exposure model, $Il36r^{-/-}$ mice and littermate WT control C57BL/6J mice were CS exposed 5 days per week with the experimental readout on day 19. Mice were exposed to five cigarettes (Roth-Händle without filters, tar 10 mg, nicotine 1 mg, carbon monoxide 6 mg, Imperial Tobacco) per day. A semi-automatic cigarette lighter and smoke generator with an electronic timer was used to control CS exposure (Boehringer Ingelheim Pharma GmbH & Co. KG, Biberach, Germany) as previously described[64]. Eighteen hours after the last CS exposure, mice were euthanized on either day 5 or day 19 with an overdose of pentobarbital (400 mg/kg intraperitoneal (i.p.)) and BAL was performed.

**CS exposure and H1N1 infection model.** $Il36r^{-/-}$ and $Il1rap^{-/-}$ mice and littermate WT control C57BL/6J mice received RA or were infected with Influenza virus in combination with CS exposure for 2 weeks as described in the 2-week CS exposure model in the section "CS exposure model." For viral infection, mice were anesthetized with 3% isoflurane and infected 2 h after smoke exposure by administering 3 Infections Units (IU) of H1N1 for $Il1rap^{-/-}$ and WT controls and 30 IU of H1N1 for $Il36r^{-/-}$ mice and WT controls in PBS in a total volume of 50 µL intranasally, 25 µL per nostril on day 7. Influenza virus A/PR/8/34 (H1N1) was obtained from Boehringer Ingelheim in Laval, Canada.

**Lung homogenate.** Lungs were removed and washed with PBS and snap frozen in liquid nitrogen immediately after extraction and weighing, and they were stored at −80 °C. For homogenization, lungs were thawed and a FastPrep-24 Sample Preparation System following the manufacturer's instructions (MP Biomedicals, Irvine, CA, USA) was used. Homogenates were then centrifuged for 10 min at $300 \times g$ and 4 °C and the supernatant was stored at −20 °C. Cytokine amounts in lung homogenate were measured by using MSD multiplex technology (Meso Scale Discovery, Gaithersburg, MD, USA) according to the manufacturer's instructions.

**Murine bone marrow-derived neutrophil isolation and stimulation.** Neutrophils were isolated from the suspension of bone marrow of male C57BL/6J mice (8–13 weeks) according to the manufacturer's instructions (Miltenyi Biotec, Neutrophil Isolation Kit mouse; #130-097-658). The neutrophils were resuspended and plated at $5 \times 10^5$ cells/mL/well in a 24-well plate in macrophage media without MCSF and stimulated with Salmonella LPS, IL-36α, IL-36β, IL-36γ, IL-1α, IL-1β, GM-CSF, and poly(I:C) (LMW) in vitro for 4 h. Cells were cultured in an incubator at 37 °C at 5% $CO_2$.

**Bronchoalveolar lavage.** Mice were euthanized by i.p. administration of an overdose of pentobarbital (400–800 mg/kg). The trachea was cannulated and BAL performed by flushing lungs twice with 0.8 mL of lavage buffer (Hanks' balanced salt solution (HBSS) containing 0.6 mM EDTA). Cell counts per mL BAL fluid (neutrophils, AMs) were measured and differentiated using the Sysmex XT-1800i automated hematology analyzer as described previously[63,64]. The BAL fluid was centrifuged for 5 min at $200 \times g$ at 4 °C. Supernatant was decanted and cytokines were measured using MSD (Meso Scale Discovery, Gaithersburg, MD, USA) or enzyme-linked immunosorbent assays (ELISAs) following the manufacturers' instructions. BAL pellet was lysed using RLT buffer (350 µL, #1053393, Qiagen) to isolate RNA for downstream cDNA generation and gene expression analyses by qPCR.

**AM isolation and stimulation.** BAL pellet was isolated as described in the section "Bronchoalveolar lavage" and resuspended and $2.5 \times 10^5$ cells/0.5 mL/well were plated in a 24-well plate in macrophage media with GM-CSF and stimulated with Salmonella LPS (4 h), IL-36α, IL-36β, IL-36γ, IL-1α, IL-1β, TNF-α, and poly(I:C) (LMW) in vitro. Cells were cultured in an incubator at 37 °C at 5% $CO_2$.

**Bone marrow-derived macrophages.** C57BL/6J mice (male, 8–13 weeks) were sacrificed by i.p. administration of an overdose of pentobarbital (400–800 mg/kg), and bone marrow cells were harvested from femurs and tibiae by flushing the respective bone with PBS. Isolated cells were centrifuged at $200 \times g$ for 5 min, supernatant was decanted, and pellets were resuspended in 100 mL of macrophage media. Cells were differentiated to macrophages in an incubator at 37 °C at 5% $CO_2$ for 6–7 days before scraping, counting, and plating for downstream experiments.

**Stimulation of BMDMs.** In all experiments, $4 \times 10^5$ macrophages/mL were seeded in a 12-well plate in macrophage media. Macrophages were allowed to adhere for 16–24 h before stimulating for 24 h with Salmonella LPS (4 h), IL-36α, IL-36β, IL-36γ, IL-1α, IL-1β, TNF-α, and poly(I:C) (LMW) in vitro. Cells were cultured in an incubator at 37 °C at 5% $CO_2$. To mimic the lung milieu, BMDMs were stimulated with TGF-β and GM-CSF in combination with IL-36α, IL-36β, IL-36γ, IL-1α, and IL-1β for 24 h.

**Fibroblast isolation and stimulation**. C57BL/6J mice (male, 8–13 weeks) were sacrificed by i.p. administration of an overdose of pentobarbital (400–800 mg/kg), and the lungs were removed and washed by dipping into ice-cold PBS. The lungs were minced and transferred into a 50-mL tube containing 10 mL DMEM (DMEM (1×)+GlutaMAXTM-I (GIBCO #31966-021), 10% FCS (GIBCO #16140-071), 1% Antibiotic Antimycotic solution (Life Technologies, cat.no. 15240062), and 100 μL of liberase solution (26 U/5 mg/mL; Roche, cat.no. 05401119001). After 2 h shaking at 37 °C, the lung pieces were strained (70 μM, BD, #352350) and flushed twice with 5 mL PBS. The tissue solution was then centrifuged for 5 min at $190 \times g$. The pellet was resuspended and washed twice in 10 mL PBS (centrifugation for 5 min at $190 \times g$). The pellet was then resuspended in DMEM (DMEM (1×)+Gluta-MAXTM-I (GIBCO #31966-021); 10% FCS (GIBCO #16140-071); 1% Antibiotic Antimycotic solution (Life Technologies, cat.no. 15240062), and transferred into a 75-cm² flask and cultured in an incubator at 37 °C at 5% $CO_2$. After the first 24 h, medium was changed. Every 2 days, the fibroblasts were washed with PBS and medium was changed until a confluence of 70% was reached to transfer the fibroblasts into a 175-cm² flask. In all, $5 \times 10^4$ primary mouse fibroblasts were seeded in a 12-well plate in macrophage media without MCSF. After 16–24 h, fibroblasts were simulated for 24 h with IL-36α, IL-36β, IL-36γ, IL-1α, IL-1β, or poly(I:C) (LMW) in vitro. Cells were cultured in an incubator at 37 °C at 5% $CO_2$.

**Stimulation of human epithelial basal cells**. In all experiments, $3 \times 10^4$ small airway epithelial cells from a human donor obtained from Lonza were seeded in a 96-well plate in 200 μL PneumaCult™-Ex Plus Medium (Stem cell Technologies, # 05040 Kit, 05041 Basal Medium). After 24 h, epithelial cells were simulated for 24 h with Salmonella LPS, IL-36α, IL-36β, IL-36γ, GMCSF, and poly(I:C) (LMW) IL-1α and IL-1β. Cells were cultured in an incubator at 37 °C at 5% $CO_2$.

**Generation of human MDMs**. Monocytes were isolated from peripheral blood mononuclear cells from human volunteer donors by negative magnetic separation using the Monocyte Isolation Kit II (MACS Miltenyi #130-091-153, Miltenyi Biotech, Bergisch Gladbach, Germany) and the AutoMACS pro system (Miltenyi Biotech, Bergisch Gladbach, Germany) according to the manufacturer's instructions. In order to generate MDMs, monocytes were resuspended in 100 mL of macrophage media. Cells were cultured in an incubator at 37 °C at 5% $CO_2$ for 6–7 days before using for experiments. All experiments with blood from human volunteer donors were approved by the blood donation service from Boehringer Ingelheim Pharma GmbH and Co. KG following ethical standards and local regulation.

**Stimulation of human MDMs**. In all experiments, $5 \times 10^5$ macrophages were seeded in a 12-well plate in macrophage media. After 16–24 h macrophages were simulated for 24 h with Salmonella LPS, IL-36α, IL-36β, IL-36γ, GM-CSF, and poly (I:C) (LMW). Cells were cultured in an incubator at 37 °C at 5% $CO_2$.

**Neutrophil isolation and stimulation**. Neutrophils were isolated from whole blood from volunteer human donors by negative magnetic separation using the MACSexpress Whole Blood Neutrophil Isolation Kit (#130-104-434, Miltenyi Biotech, Bergisch Gladbach, Germany) according to the manufacturer's instructions. In all, $3–5 \times 10^5$ neutrophils were seeded in a 24-well plate in macrophage media without MCSF. All experiments with blood from human volunteer donors were approved by the blood donation service from Boehringer Ingelheim Pharma GmbH and Co KG following ethical standards and local regulation.

**Stimulation of primary human fibroblasts**. In all experiments, $2 \times 10^4$ normal human lung fibroblasts obtained from Lonza were seeded in a 12-well plate in macrophage media without MCFS and simulated for 24 h with Salmonella LPS, IL-36α, IL-36β, IL-36γ, GM-CSF, and poly(I:C) (LMW). Cells were cultured in an incubator at 37 °C at 5% $CO_2$.

**Single-cell RNA sequencing**. The 10× Genomics Chromium system and Single Cell 3' Reagent v2 Kits (10× Genomics, Pleasanton, CA) were used to generate scRNA-seq libraries according to the manufacturer's instructions. The reverse transcribed cDNA was first amplified by 14 PCR cycles and then used to generate the sequencing library, which was finally amplified by 14 PCR cycles. The sequencing library was purified as described by the manufacturer and then sub-jected to another cleanup step using one volume of SPRISelect Beads (Beckman Coulter, Brea, CA, USA) in order to remove leftover primers and dimers. The average length of the purified library was 494 bp. A HiSeq 4000 and two 50-cycle SBS kits (Illumina, San Diego, CA, USA) were used for sequencing and clusters were generated on a cBot using the HiSeq 3000/4000 PE Cluster Kit. The paired-end sequencing run comprised a 26-bp read 1 (16 bp cell barcode and 10 bp UMI) and 98-bp read 2 (transcript sequence read) as well as a 8-bp index read. On average, approximately 50,000 reads were sequenced per cell.

**Data processing and analysis**. Raw data were processed using the Cell Ranger v2.1.1[65] pipeline to finally obtain a count matrix that was used as input for downstream analysis by the Seurat R package v3.0.0.9[66]. During data processing,

reads were aligned to the GRCm38 reference genome and annotated according to Ensembl release 86[67].

For downstream analysis, low quality cells with <300 detected genes and/or >10% of mitochondrial transcripts were removed from the data set. Additionally, cells with >5000 detected genes and genes that were detected in less than three cells were removed from the data set, which left 3075 cells and 14,728 genes for further analysis.

Data were then normalized and log-transformed per cell using the "NormalizeData" function. In order to identify cell clusters within the data set, the highly variable genes were first calculated by the "FindVariableFeatures" function and "mean.var.plot" method. Data were then scaled and centered and principal component (PC) analysis was computed on the variable genes using the "RunPCA" function. Cell clusters were then calculated using the first 13 PCs as input and a resolution of 0.2 and visualized by UMAP[68] representation that was also calculated on the first 13 PCs.

Cell clusters were annotated based on the expression of characteristic marker genes. A small cluster that represented 3.4% of the cells was characterized by the expression of both macrophage and neutrophil marker genes (Suppl. Fig. 6b, c) as well as a low UMI count that was 12- and 2.3-fold lower than for macrophages and neutrophils, respectively. Therefore, this cluster was identified as a low-quality cluster and removed for further analysis. Subcluster analysis of the neutrophils was performed as described for the entire data set using only the neutrophils. Neutrophil clusters were computed on the first 10 PCs and a resolution of 0.2. The first ten PCs were also used for UMAP dimensional reduction.

**Differential expression analysis and gene set enrichment analysis**. Differential expression analysis was calculated by the MAST R package v1.6.1[69] via Seurat's "FindMarkers" function. Genes were kept for differential expression analysis if they were detected in at least 10% of the cells of the compared groups and considered to be significantly differentially expressed if they were at least 1.5-fold differentially expressed with a $P$ value adjusted for multiple testing <0.05 (Bonferroni correction). Gene set enrichment analysis was performed for the significantly increased genes per neutrophil cluster using the g:Profiler webtool[70] and the default settings data sources.

**LPS exposure for scRNA-seq study**. Female C57BL/6JCR mice (18–20 g, Charles River Research Models and Services Germany GmbH, Sulzfeld, Germany) were used for the acute LPS model that was used for the scRNA-seq study. Mice were exposed to LPS as described in the section "LPS aerosol exposure" and euthanized 4 h post LPS inhalation by i.p. pentobarbital injection (Narcoren, Merial GmbH, Halbermoos, Germany). Mice were given ad libitum access to water and food during the experiment.

**BAL and cell purification for single-cell study**. BAL was performed four times per animal using 0.8 mL of HBSS per lavage per animal. For scRNA-seq analysis, BAL fluids of eight animals were equally pooled to obtain a single BAL pool that contained the same cell count per animal. BAL cells were centrifuged at $400 \times g$ and 4 °C for 5 min and the supernatant was removed. The cell pellet was washed once using 10 mL of HBSS/0.04% bovine serum albumin (BSA) buffer and finally resuspended in 500 μL of HBSS/0.04% BSA buffer and filtered through a 40-μm cell strainer.

**RNA isolation and qPCR**. Cells were washed with PBS and lysed using 350 μL RLT buffer (350 μL, Qiagen #1053393). RNA was purified using MagMax (Biosystems #AM1830) following the manufacturer's instructions. RNA was transcribed to cDNA using the High Capacity cDNA Reverse Transcription Kit (Biosystems #4368813) according to the manufacturer's instructions. Taqman master mix (Applied Biosystems #4352042) was used to prepare the RT-qPCR reaction mix. Taqman gene expression assays were obtained from ThermoFischer Scientific: Housekeeper Hprt1: Mm03024075_m1 or 18S: Mm03928990_g1; IL1F9: Mm00463327_m1; MMP-9: Mm00463327_m1; CXCL1: Mm04207460_m1; IL1β: Mm00434228_m1; IL1F6: Mm00457645_m1; IL1F8: Mm01337546_g1;Il1a: Mm00439620_m1; Il6: Mm00446190_m1; IL1RL2: Mm00519245_m1. Plates were prepared following the manufacturer's instructions and analyzed on an Applied Biosystems QuantStudio 6 Flex Real-Time PCR System. Gene expression was normalized to the housekeeper (dCt) and depicted as $2^{(ddCt)}$.

**Cytokine measurements**. To measure the protein concentrations in supernatant, MSDs (Meso Scale Discovery, Gaithersburg, MD, USA; V-Plex Proinflammatory Panel 1 mouse Kit; K15048D-1; and ELISAs such as the CXCL1/KC DuoSet ELISA (R&D Systems, #DY453-05) and the mouse IL-36γ ELISA (Aviva Systems Biology, #OKEH03002) were used according to the manufacturer's instructions.

**Statistical analyses and reproducibility**. Data were analyzed by using the Prism 8 software package (GraphPad Software, San Diego, CA). All data were expressed as mean and standard error of the mean (SEM) and were analyzed using Student's $t$ test or one-way analysis of variance with a value of <0.05 considered to be sta-tistically significance. The reproducibility was determined by using several

biological replicates and repeating the experiment several times as indicated in the figure legends.

**Reporting summary**. Further information on research design is available in the Nature Research Reporting Summary linked to this article.

## Data availability

All data are available from the corresponding author upon request and the data related to the RNA-seq experiment are deposited in GEO reference GSE159161. Source data underlying the main and supplementary figures are available in Supplementary Data 1.

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

## Acknowledgements

We are grateful to Dr. Peter Murray for his critical appraisal of the manuscript. We acknowledge Dr. David Lamb for his in vivo model expertise. We thank Jochen Blender for help with the technical in vivo aspects.

## Author contributions

K.C.E.K.: conceived of the study, designed the study, interpreted the data, and wrote the manuscript; C.K.K.: conceived and designed the research study, conducted the bulk of the experiments, interpreted the data, and wrote the manuscript; P.J.B., L.E.D., J.R.B.: contributed to interpreting the data; C.T.W.: conducted the bulk of the single cell experiment; C.T., C.L., M.P., S.F., M.K.: contributed to the experimental execution of the experiments; C.M.M.W., D.P., M.R., J.F., F.G., M.T.: contributed to the conception and design of the study; all authors contributed to scientific discussions, revision of the manuscript, had full access to all the data, and agreed to submit for publication.

## Competing interests

The authors declare no competing interests.
