## [Peer Review File · Communications Biology]

Reviewers' comments:

Reviewer #1 (Remarks to the Author):

Koss et. al. reported that IL-36, a member of IL-1 family, amplifies neutrophilic lung inflammation. IL-36 cytokines have been implied in skin inflammation, arthritis, and intestinal inflammation. In the lung, IL-36 has been suggested to be important in the pathogenesis of experimental bacterial and viral pneumonia in mice. There are also several human lung diseases linked to IL-36 cytokine(s) to neutrophilic inflammation in multiple diseased tissues; however, the underlying mechanisms are unknown. This study combined chronic and acute lung inflammation mouse models with in vivo and in vitro approaches using primary mouse and human cells. With the well-designed experiments and genetic tools, they found that neutrophils are a source of IL-36 and IL-36 is a key upstream amplifier of lung inflammation by promoting activation of neutrophils, macrophages, and fibroblasts through cooperation with GM-CSF and the viral mimic poly(I:C). The study provides novel mechanistic insight that identifies IL-36 within the IL-1 family of cytokines as an early upstream innate immune driver and amplifier of acute and chronic lung inflammation.

The manuscript is well written and organized. There are several minor concerns, nonetheless, they don't affect the major findings of this study.

1. Poly(I:C) and LPS were used to test the effect on cytokine expression. Have the authors examined the levels of type I IFN as they are induced directly by these ligands?
2. In the introduction, it is not clear what types of cells express the IL-36 receptor.
3. Line 74, 194-196. The format of the numbers needs to change.

Reviewer #2 (Remarks to the Author):

The authors used novel in vivo models to show the relevance of IL36 during lung inflammation. They thoroughly showed the roles and the complex interplay of alveolar macrophages, neutrophils and fibroblasts in their function as producers of IL36 or responders to IL36 during lung inflammation. The results are summed up in a complex model (figure 10). The novelty regarding target genes is limited as similar mechanisms are described in the skin and the intestine. To increase the novelty, the author could further focus on the role of tissue remodeling during lung diseases as they mentioned MMP9 as target gene of IL36R signaling in suppl. Figure 3C. However, the results of Kasmi et al. confirm the fact, that IL36R signaling is crucial in proinflammatory signaling in organs with direct contact to the environment. It is also important to mention, that the authors showed a mechanism of proinflammatory IL36R signaling independent of IL1R signaling. As a consequence, a neutralization of IL36R signaling as therapy option does not affect IL1 mediated host defense. The conclusions could be further strengthened by confirmation in human lung diseases. The statistical analysis is valid and the materials and methods section provide thorough information about the performed experiments and bioinformatics analysis. The results from murine in vitro stimulation is consistent with results from human cells. In order to improve the manuscript for publication the authors are kindly asked to provide answers to the following issues:

Issues:

- The amount of human data is limited to human in vitro experiment with human cells. Data about the relevance of IL36 for human inflammatory lung diseases is missing. Based on that fact, the authors should tune down the title as it indicates also a relevance in human diseases. Or add more human data.
- Figure 10 proposed model: The model is quite complex and reflects the results of the paper. To improve the understandability of the interplay between neutrophils, macrophages and fibroblasts the authors should reduce the model to the key results. It is also possible to add numbers to understand the interplay in a timely manner started with the trigger of inflammation.
- The proposed model in fig. 10 is not mentioned in the results or the discussion section. The

author should include some sentences in the results.

- L442: Exact description of used mice: IL36 or IL36R ko?
- Please use a consistent terminology throughout the manuscript: IL-36R or Il-36r e.g L502, L507, Fig 5 and the corresponding legend etc.
- Supplementary figure 2, 7, 8 are missing
- Figure 3E: the stimulation conditions are not clear as well as the analyzed genes (two times IL36r?)
- Suppl figure 3C: Is MMP9 also relevant in the mouse models in vivo? Or human disease?
- The headlines in the results section are quite redundant. The authors should try to specify/sum up the results from each figure in the headline. In addition, there is often no summary of the results at the end of the results section.
- How long where the mice exposed to CS in fig 5 B, C, D, E, F, G? Please include this information in the figure legend.
- The inflammatory phenotype of IL36Rko attenuated. The neutralization of IL36R signaling in vivo could be a therapy option in inflammatory lung diseases. However, in case of infections, would the blockade of inflammatory responses be a risk for clearance of the pathogen?
- Fig. 5E-G: Is there also a difference in the phenotype of IL36Rko animals of the CS and the viral infection model compared to virus+CS? The authors should show the results at least in the supplemental figures.
- The use of the different mouse models is insufficient explained. Why do the author introduce a new model for the scRNAseq? The results from the previous described models were promising.
- The authors should specify the role of IL36 in each model in more detail as one could imagine that cigarette smoke, bacterial and viral compounds can have differential effect on lung cells as well as there a different effects of acute and chronic models. Of note, the 3 triggers of lung inflammation do probably affect humans simultaneously. However, the advantage of separate models can help to understand the response of lung cells to IL36 under the specific conditions.
- The results of the scRNAseq confirmed previous results and showed that IL36g is the main IL36R ligand expressed in lung inflammation. This specialized method bears the opportunity to analyze more than just the identification of neutrophils as source of IL36g. Which signaling pathways are increased? What is the expression of MMP9 or other genes involved in tissue remodeling? Etc. The authors should include more results of the scRNAseq in order to emphasize the role of IL36 in lung inflammation and tissue remodeling.
- L650: the described information is not in suppl fig 1a,b, please correct the reference
- Figures: the order of the subfigures is rather confusing (vertical), please improve the structure of figure 1, 2, 3, 5, 9 as you did in the supplemental figures (horizontal)
- Are AM also increased in the analyzed models? Corresponding to fig 1A. This fact should be shown as downstream analysis of the effect of IL36 on AM is rather not relevant.

Response to Reviewers:

Point-by point response to Reviewer #1

Remarks to the Author:

Koss et. al. reported that IL-36, a member of IL-1 family, amplifies neutrophilic lung inflammation. IL-36 cytokines have been implied in skin inflammation, arthritis, and intestinal inflammation. In the lung, IL-36 has been suggested to be important in the pathogenesis of experimental bacterial and viral pneumonia in mice. There are also several human lung diseases linked to IL-36 cytokine(s) to neutrophilic inflammation in multiple diseased tissues; however, the underlying mechanisms are unknown. This study combined chronic and acute lung inflammation mouse models with in vivo and in vitro approaches using primary mouse and human cells. With the well-designed experiments and genetic tools, they found that neutrophils are a source of IL-36 and IL-36 is a key upstream amplifier of lung inflammation by promoting activation of neutrophils, macrophages, and fibroblasts through cooperation with GM-CSF and the viral mimic poly(I:C). The study provides novel mechanistic insight that identifies IL-36 within the IL-1 family of cytokines as an early upstream innate immune driver and amplifier of acute and chronic lung inflammation.

The manuscript is well written and organized. There are several minor concerns, nonetheless, they don't affect the major findings of this study.

Response to reviewer #1: *We thank the reviewer for the very positive appraisal of our study and for pointing out that the manuscript is well written and well organized. We appreciate the reviewer's comments that the study was well-designed and that it provides novel mechanistic insight.*

We have provided a response to the minor points raised by the reviewer below:

Point 1. Ploy(I:C) and LPS were used to test the effect on cytokine expression. Have the authors examined the levels of type I IFN as they are induced directly by these ligands?

*Thank you for this comment, we recognize that determining type IFN expression downstream of IL36 provides valuable information. We therefore determined the IFN β 1 mRNA expression in naïve mouse bone marrow derived macrophages and naïve primary lung fibroblasts in response to IL36 and poly IC. We found that, as expected, poly IC induced significant increases in type I IFN expression in BMDMs. In contrast, IL36 stimulation did not promote type I IFN expression. Moreover, the combination of IL36 and Poly IC resulted in some suppression of type I IFN expression relative to poly IC alone. Fibroblasts did not express IFN β 1 after PolyI:C or IL36 stimulation. We therefore conclude that blocking IL36 signaling will likely not interfere with the ability to generate antiviral interferon responses. We have included these data in the manuscript in **Supplementary Figure 6G** and included a statement in the "**results**" (line 266-269). Furthermore, we provide a more detailed analysis of our scRNAseq experiment and show that while IFN β 1 itself is not expressed by neutrophils we do find IFN pathway induction. The latter finding is now included in **Supplementary Figure 9G** (line 327-333).*

Supplementary Fig 6 IL-36 β cooperates with poly(I:C) on macrophages and fibroblasts a-b Neutrophils and macrophage numbers in BALF and GM-CSF, CXCL1 and MMP9 protein concentrations in BALF and relative *IL36* mRNA amounts in the BAL cell pellet of untreated (n=4) and poly(I:C) exposed mice (n=8) for the indicated amounts of time. c Relative mRNA amounts of *IL36*, *Cxcl1*, *Ifi1*, *Ifi2* and *IL-36 in alveolar macrophages stimulated with poly(I:C) *in vitro* (AMs were pooled from 4 naive). d-f CXCL1, IL-36 β , GM-CSF and MMP9 protein concentrations in supernatants from BMDM (n=2) and primary mouse lung fibroblasts (n=4) either unstimulated (-) or stimulated with IL-36 β , poly(I:C) or the combination of IL-36 β and poly(I:C). e Relative mRNA amounts of *IL36* in BMDM (n=4) and primary mouse lung fibroblasts (n=4) as well as after poly(I:C) stimulation. g Relative mRNA amounts of *Inf1* in BMDMs (n=4) and primary mouse lung fibroblasts (n=4) and BMDMs (n=4) after IL-36 β , poly(I:C) or the combination of IL-36 β and poly(I:C) relative to no stimulation (-). (Depicted are mean values \pm SEM of biological replicates). (a,b,d,f,g) *P < 0.05, **P < 0.01, ***P < 0.001, ****P < 0.0001 vs all other groups by one-way ANOVA and Tukey's correction. (c,e) *P < 0.05, **P < 0.01, ***P < 0.001 vs untreated by t test.*

Supplementary Fig 9 Neutrophils are a source of IL-36 β in acute lung injury. a Neutrophil numbers in BAL samples from untreated and 2, 4, 6, 8, 12, 18, 24, 36, 48 and 60 hours post LPS exposure (n=8). b UMAP representation of the cell clusters (left) and cell populations (right) identified in BAL of mice 4 h after LPS stimulation. c Expression of the marker genes used to annotate the cell types. Data are shown per cell type using the color code used in (a). Scaled average expression levels are indicated by the dot color and dot size indicates the fraction of cells that express the respective marker gene. d Visualization of *IL36a* (*Ifi16*) expression levels per single cell. Normalized expression values are indicated by the color code. e *IL-36a*, *IL-36b* and *IL-36g* mRNA expression of alveolar macrophages unstimulated (-), or exposed to LPS for 4 h (Depicted are mean values \pm SEM of biological replicates). f UMAP representing the subclusters (N1 and N2) identified within the neutrophils from (b). g Gene set enrichment analysis (GSEA) of the neutrophil subclusters N1 and N2. Selected pathways are shown per subcluster. h Visualization of *Mmp9* expression in neutrophils. Normalized expression levels per single cell are indicated by the color code.

Point 2: In the introduction, it is not clear what types of cells express the IL-36 receptor.

We have included in the introduction which cell types are described to express IL-36R (line 57,58).

Point 3. Line 74, 194-196. The format of the numbers needs to change. ???

We have changed the format (line 75, 201-203).

Point-by point response to Reviewer #2

Remarks to the Author:

The authors used novel *in vivo* models to show the relevance of IL36 during lung inflammation. They thoroughly showed the roles and the complex interplay of alveolar macrophages, neutrophils and fibroblasts in their function as producers of IL36 or responders to IL36 during lung inflammation. The results are summed up in a complex model (figure 10). The novelty regarding target genes is limited as similar mechanisms are described in the skin and the intestine. To increase the novelty, the author could further focus on the role of tissue remodeling during lung diseases as they mentioned MMP9 as target gene of IL36R signaling in suppl. Figure 3C. However, the results of Kasmir et al. confirm the fact, that IL36R signaling is crucial in proinflammatory signaling in organs with direct contact to the environment. It is also important to mention, that the authors showed a mechanism of proinflammatory IL36R signaling independent of IL1R signaling. As a consequence, a neutralization of IL36R signaling as therapy option does not affect IL1 mediated host defense. The conclusions could be further strengthened by confirmation in human lung diseases. The statistical analysis is valid and the materials and methods section provide thorough information about the performed experiments and bioinformatics analysis. The results from murine *in vitro* stimulation is consistent

with results from human cells. In order to improve the manuscript for publication the authors are kindly asked to provide answers to the following issues:

We thank the reviewer for the positive appraisal of our manuscript and for emphasizing that we have mechanistically elucidated a complex interplay of alveolar macrophages, neutrophils and fibroblasts. We also appreciate the recognition that “neutralization of IL36R signaling as therapy option does not affect IL1 mediated host defense”. We furthermore appreciate your helpful comments, which we have addressed in a point-by-point response below.

Point #1: The amount of human data is limited to human in vitro experiment with human cells. Data about the relevance of IL36 for human inflammatory lung diseases is missing. Based on that fact, the authors should tune down the title as it indicates also a relevance in human diseases. Or add more human data.

We agree with the reviewer and we have changed the title to better reflect the lack of human data as follows: “IL36 is a Critical Upstream Amplifier of Neutrophilic Lung Inflammation in Mice”.

Point # 2: Figure 10 proposed model: The model is quite complex and reflects the results of the paper. To improve the understandability of the interplay between neutrophils, macrophages and fibroblasts the authors should reduce the model to the key results. It is also possible to add numbers to understand the interplay in a timely manner started with the trigger of inflammation.

We agree with the reviewer and we have simplified it and increased the visual attractiveness of the cartoon (Figure 10).

Fig. 10 Proposed signaling pathways involving IL-36 as an upstream amplifier in neutrophilic lung inflammation. 1 Chronic (e.g. cigarette smoke) or acute (H1N1 or poly(I:C)) injury to the lung epithelium promotes release of CXCL1 and GM-CSF. 2 CXCL1 promotes neutrophil recruitment while GM-CSF up-regulates the IL-36R and IL1RN on neutrophils and promotes IL-36y expression. 3 IL-36y activates neutrophils to generate IL-1 and IL-36. 4 IL-36y subsequently activates alveolar and interstitial macrophages and fibroblasts to generate IL-1, GM-CSF, CXCL1, MMP9 and more IL-36. GM-CSF also up-regulates the IL36R on macrophages and on fibroblasts and sensitizes cells to IL-36. 5 Poly(I:C) as TLR agonist cooperates with IL-36y in amplifying activation of macrophages and fibroblasts. Note that IL-36RN is not induced while IL1RN is induced, shifting the balance towards heightened responsiveness to IL-36.

Point # 3: The proposed model in fig. 10 is not mentioned in the results or the discussion section. The author should include some sentences in the results.

We now refer to Fig. 10 in the “results” (line 334-335) and “discussion” (line 452-458).

Point # 4: Suppl figure 3C: Is MMP9 also relevant in the mouse models in vivo? Or human disease?

*While we agree with the reviewer that MMP9 in human disease and in vivo models is of considerable interest for a better understanding of the pathophysiology of lung tissue remodeling, we consider addressing this question in greater detail is beyond the scope of the present manuscript. However, in order add to our initial data presented in the manuscript we now include **new data** that show increased expression of MMP9 after Poly(I:C) treatment in vivo (Suppl. Figure 6A) (line 242,247)*

and MMP9 protein and mRNA increase after Poly(I:C) and IL36 stimulation (Suppl. Figure 6D,F) (line.260). We have also found increased MMP9 expression in the neutrophils (subcluster 1) in the LPS in vivo scRNA experiment (Suppl. Figure 9H) (line 327-333). To test the relevance of MMP9 as a remodeling factor in vivo more rigorously, additional, more complex experiments using longer term animal models would need to be conducted to address a physiological role of MMP9 downstream of IL36 in tissue remodeling. We consider these aspects beyond the scope of the present manuscript.

Supplementary Fig 6 IL-36 cooperates with poly(I:C) on macrophages and fibroblasts a-b Neutrophils and macrophage numbers in BAL and GM-CSF, CXCL1 and MMP9 protein concentrations in BALF and relative *I36g* mRNA amounts in the BAL cell pellet of untreated (n=4) and poly(I:C) exposed mice (n=8) for the indicated amounts of time. c Relative mRNA amounts of *I36g*, *Cxcl1*, *Ifa*, *Ifb* and *Il-36r* in alveolar macrophages stimulated with poly(I:C) *in vitro* (AMs were pooled from 4 naïve). d-f CXCL, IL-36, GM-CSF and MMP9 protein concentrations in supernatants from BMDM (n=2) and primary mouse lung fibroblasts (n=4) either unstimulated (-) or stimulated with IL-36α, poly(I:C) or the combination of IL-36α and poly(I:C). e Relative mRNA amounts of *I36r* in BMDM (n=4) and primary mouse lung fibroblasts (n=4) as well as after poly(I:C) stimulation. g Relative mRNA amounts of *Ifnb1* in BMDMs (n=4) and primary mouse lung fibroblasts (n=4) and BMDMs (n=4) after IL-36α, poly(I:C) or the combination of IL-36α and poly(I:C) relative to no stimulation (-). (Depicted are mean values ± SEM of biological replicates). (a,b,d,f,g) *P < 0.05, **P < 0.01, ***P < 0.001, ****P < 0.0001 vs all other groups by one-way ANOVA and Tukey's correction. (c,e) *P < 0.05, **P < 0.01, ***P < 0.001, ****P < 0.0001 vs untreated by t test.

Supplementary Fig 9 Neutrophils are a source of IL-36 in acute lung injury. a Neutrophil numbers in BAL samples from untreated and 2, 4, 6, 8, 12, 18, 24, 36, 48 and 60 hours post LPS exposure (n=8). b UMAP representation of the cell clusters (left) and cell populations (right) identified in BAL of mice 4 h after LPS stimulation. c Expression of the marker genes used to annotate the cell types. Data are shown per cell type using the color code used in (a). Scaled average expression levels are indicated by the dot color and dot size indicates the fraction of cells that express the respective marker gene. d Visualization of *I36a* (*Ifb*) expression levels per single cell. Normalized expression values are indicated by the color code. e *Il-36a*, *Il-36b* and *Il-36g* mRNA expression of alveolar macrophages unstimulated (-), or exposed to LPS for 4 h (Depicted are mean values ± SEM of biological replicates). f UMAP representing the subclusters (N1 and N2) identified within the neutrophils from (b). g Gene set enrichment analysis (GSEA) of the neutrophil subclusters N1 and N2. Selected pathways are shown per subcluster. h Visualization of *Mmp9* expression in neutrophils. Normalized expression levels per single cell are indicated by the color code.

Point # 4: The inflammatory phenotype of IL36Rko is attenuated. The neutralization of IL36R signaling in vivo could be a therapy option in inflammatory lung diseases. However, in case of infections, would the blockade of inflammatory responses be a risk for clearance of the pathogen?

The reviewer raises an important point here. We believe that our study addresses this point by showing that the mechanism of proinflammatory IL36R signaling is independent of IL1R signaling. As a consequence, neutralization of IL36R signaling as a therapy option is not expected to affect IL1 mediated host defense. We have included a statement in the discussion to elaborate on this important point raised by the reviewer. Furthermore, we determined IFN β 1 mRNA expression in naïve mouse bone marrow derived macrophages and naïve primary lung fibroblasts in response to IL36 and poly IC. We found that, as expected, poly IC induced significant increases in type I IFN expression. In contrast, IL36 stimulation did not promote type I IFN expression. Moreover, the combination of IL36 and Poly IC resulted in some suppression of type I IFN expression relative to poly IC alone. We therefore conclude that blocking IL36 signaling will likely not interfere with the ability to generate antiviral interferon responses. We have included these data in the manuscript in **Supplementary Figure 6G** (line 266-269).

Supplementary Fig 6 IL-36 γ cooperates with poly(I:C) on macrophages and fibroblasts a-b Neutrophils and macrophage numbers in BAL and GM-CSF, CXCL1 and MMP9 protein concentrations in BALF and relative #36g mRNA amounts in the BAL cell pellet of untreated (n=4) and poly(I:C) exposed mice (n=8) for the indicated amounts of time. c Relative mRNA amounts of #36g, Cxcl1, #1a, #1b and #36r in alveolar macrophages stimulated with poly(I:C) *in vitro* (AMs were pooled from 4 naïve). d-f CXCL1, IL-36 γ , GM-CSF and MMP9 protein concentrations in supernatants from BMDM (n=2) and primary mouse lung fibroblasts (n=4) either unstimulated (-) or stimulated with IL-36 γ , poly(I:C) or the combination of IL-36 γ and poly(I:C). e Relative mRNA amounts of #36r in BMDM (n=4) and primary mouse lung fibroblasts (n=4) as well as after poly(I:C) stimulation. g Relative mRNA amounts of #1b1 in BMDMs (n=4) and primary mouse lung fibroblasts (n=4) and BMDMs (n=4) after IL-36 γ , poly(I:C) or the combination of IL-36 γ and poly(I:C) relative to no stimulation (-) (Depicted are mean values \pm SEM of biological replicates). (a,b,d,f,g) *P \leq 0.05, **P \leq 0.01, ***P \leq 0.001, ****P \leq 0.0001 vs all other groups by one-way ANOVA and Tukey's correction. (c,e) *P \leq 0.05, **P \leq 0.01, ***P \leq 0.001, ****P \leq 0.0001 vs untreated by t test.

Point # 5: Fig. 5E-G: Is there also a difference in the phenotype of IL36Rko animals of the CS and the viral infection model compared to virus+CS? The authors should show the results at least in the supplemental figures.

Due to the detailed analysis of the phenotypes in the IL1RaP ko model we focused on the comparison of phenotypes between wild type and IL36ko in the CS+Virus model, which is in our estimation the most disease relevant and mechanistic model and therefore did not include a detailed comparison of the phenotypes of CS IL36R ko animals and the viral infection model with those of the virus+CS model.

Point # 6: The use of the different mouse models is insufficiently explained. Why do the authors introduce a new model for the scRNAseq? The results from the previously described models were promising.

We agree with the reviewer and would like to quote the reviewer's statement from below: "Of note, the 3 triggers of lung inflammation do probably affect humans simultaneously. However, the advantage of separate models can help to understand the response of lung cells to IL36 under the specific conditions". We thank the reviewer for this comment, and we have reflected that perspective in the results at the beginning of the section about the scRNA sequencing (line 291-293).

Point # 7: The authors should specify the role of IL36 in each model in more detail as one could imagine that cigarette smoke, bacterial and viral compounds can have differential effect on lung cells as well as there a different effects of acute and chronic models. Of note, the 3 triggers of lung inflammation do probably affect humans simultaneously. However, the advantage of separate models can help to understand the response of lung cells to IL36 under the specific conditions.

We agree with the reviewer and we specifically designed the study to employ various mouse models to address exactly this point. We have now included a statement in the results (line 291-293)

inspired by your perspective offered above “advantage of separate models can help to understand the response of lung cells to IL36 under the specific conditions” to reflect this better.

Point # 8: The results of the scRNAseq confirmed previous results and showed that IL36g is the main IL36R ligand expressed in lung inflammation. This specialized method bears the opportunity to analyze more than just the identification of neutrophils as source of IL36g. Which signaling pathways are increased? What is the expression of MMP9 or other genes involved in tissue remodeling? Etc. The authors should include more results of the scRNAseq in order to emphasize the role of IL36 in lung inflammation and tissue remodeling.

*Thank you for this comment. After a more detailed analysis, we have identified two distinct neutrophil subclusters. A pathway analysis revealed that one cluster expresses remodeling pathways. These new findings are now included in **Supplementary Figure 9 F,G,H** (line 327-333).*

Supplementary Fig 9 Neutrophils are a source of IL-36 γ in acute lung injury. **a** Neutrophil numbers in BAL samples from untreated and 2, 4, 6, 8, 12, 18, 24, 36, 48 and 60 hours post LPS exposure (n=8). **b** UMAP representation of the cell clusters (left) and cell populations (right) identified in BAL of mice 4 h after LPS stimulation. **c** Expression of the marker genes used to annotate the cell types. Data are shown per cell type using the color code used in (a). Scaled average expression levels are indicated by the dot color and dot size indicates the fraction of cells that express the respective marker gene. **d** Visualization of *IL36a* (#116) expression levels per single cell. Normalized expression values are indicated by the color code. **e** *IL-36a*, *IL-36b* and *IL-36g* mRNA expression of alveolar macrophages unstimulated (-), or exposed to LPS for 4 h (Depicted are mean values \pm SEM of biological replicates). **f** UMAP representing the subclusters (N1 and N2) identified within the neutrophils from (b). **g** Gene set enrichment analysis (GSEA) of the neutrophil subclusters N1 and N2. Selected pathways are shown per subcluster. **h** Visualization of *Mmp9* expression in neutrophils. Normalized expression levels per single cell are indicated by the color code.

Point # 9: Are AM also increased in the analyzed models? Corresponding to fig 1A. This fact should be shown as downstream analysis of the effect of IL36 on AM is rather not relevant.

*We thank the reviewer for bringing up the question if alveolar macrophage numbers are affected in the various models. We now include data in Fig 1A and 1B that show that AMs are not changes after CS exposure or IL36 treatment in vivo (line 75-76; 86). In addition, we provide data that show that AM numbers are 77% in the LPS model (**Figure 9A**) (line 297).*

Fig. 1 IL-36 γ as an upstream inflammatory driver in mouse neutrophils and alveolar macrophages. a Neutrophil and macrophage counts and myeloperoxidase concentration in the bronchoalveolar lavage (BAL) from room air (RA) exposed (WT n=9; *IL36 γ* ^{-/-} n=6) and 3-week cigarette smoke (CS) exposed mice (WT n=9; *IL36 γ* ^{-/-} n=10). b-d Neutrophil and macrophage counts, CXCL1, IL-1 α , IL-1 β and GM-CSF protein concentrations of the BALF from untreated (n=3) and IL-36 γ exposed mice (bronchoalveolar lavage) (n=7). e-f *Cxcl1*, *Itf1*, *Itf2*, and *Itf3* mRNA expression in either (e) naive mouse alveolar macrophages (pooled n=15 mice) and stimulated *in vitro* with either no cytokines (-), IL-30 γ , or IL-1 α /IL-1 β ; or (f) in mouse bone marrow derived neutrophils (from n=4 mice) *in vitro* stimulated with either no cytokines (-), IL-30 γ , GM-CSF, IL-30 γ +GM-CSF. (g) *Itf3* and *Itf1* mRNA expression in mouse bone marrow derived neutrophils (from n=4 mice) *in vitro* stimulated with no cytokines (-), IL-30 γ , GM-CSF, IL-30 γ +GM-CSF. (a-g) $^{*}p < 0.05$, $^{**}p < 0.01$, $^{***}p < 0.001$, $^{****}p < 0.0001$ vs untreated by *t*-test. (e-f) ANOVA and Tukey's correction. (b,c,d,g) $^{*}p < 0.05$, $^{**}p < 0.01$, $^{***}p < 0.001$, $^{****}p < 0.0001$ vs untreated by *t*-test. (g) AMs were pooled from 15 mice, data are presented as (mean \pm SEM) of technical replicates.

Fig. 2 Neutrophils are a source of IL-36 γ in acute lung injury. a UMAP representation of the cell populations identified in BAL from LPS exposed mice. Cell identity is indicated by the color code. Relative frequency of the cell types is shown in the bar plot using the same color code. b Violin plots display normalized expression levels of *Itf3* (*ITIF3*), *Itf1*, *Itf2* and *Cxcl1* across the different cell populations. c Relative mRNA amounts of the same genes used in (b) after 4h *in vitro* LPS stimulation of naive mouse alveolar macrophages (AM) and naive mouse bone marrow derived neutrophils; depicted are mean values \pm SEM of biological duplicates (each biological duplicate represents pooled AMs from 4 mice) and n=8 from the neutrophils analyzed by one-way ANOVA. d Visualization of normalized IL-36 γ (*IL36G*) expression levels per single cell. e mRNA expression of IL-36 γ after LPS stimulation *in vitro* of human neutrophils, MDMs (depicted are mean values \pm SEM of technical replicates from one representative of three to four experiments) and human fibroblasts; n=4. Data represents the depicted mean \pm SEM of biological replicates; $^{*}p < 0.05$, $^{**}p < 0.01$, $^{***}p < 0.001$, $^{****}p < 0.0001$ vs untreated by *t*-test.

Minor points:

We have corrected all the minor points as follows:

- L650: the described information is not in suppl fig 1a,b, please correct the reference

We have corrected this information (line 671).

- Figures: the order of the subfigures is rather confusing (vertical), please improve the structure of figure 1, 2, 3, 5, 9 as you did in the supplemental figures (horizontal)

We have corrected the order of the figures.

- L442: Exact description of used mice: IL36 or IL36R ko?

We have corrected this spelling mistake (line 469).

- Please use a consistent terminology throughout the manuscript: IL-36R or IL-36r e.g L502, L507, Fig 5 and the corresponding legend etc.

We have adapted the consistency of the terminology.

Fig. 5 IL-36 γ is critical in neutrophilic lung inflammation a Neutrophil counts in BALF from room air (RA) (n=18) exposed, or from 1 week (n=9), 2 week (n=9), and 3 week (n=8) cigarette smoke (CS) exposed mice, and from H1N1 (4 days post treatment) exposed mice (n=8), and from mice exposed to 2 weeks CS followed by 48hrs of H1N1 exposure (n=8). b Relative mRNA amounts of IL-36 γ in the cellular BAL pellet from room air (RA) and 2 week cigarette smoke (CS) exposed mice (line, n=number). c-d neutrophil numbers in BALF and IL-1 β , CXCL1, and IL-6 protein concentrations in lung homogenate of room air exposed mice (RA; n=4), 2 week CS exposed mice (n=4-8) and 2 week CS exposed mice challenged with H1N1 for 48hrs (CS+H1N1) (n=4-8). e Neutrophil numbers in BALF and relative mRNA amounts of IL-36 γ and Cxcr2 in lung homogenate from room air exposed (RA, n=5) and 2 week CS exposed mice challenged with H1N1 for 48hrs (n=7-8) exposed WT and IL36 γ ^{-/-} mice. f,g IL-1 β , CXCL1, IL-6 and TNF protein concentrations in BALF and in lung homogenate from room air exposed (RA, n=5) and 2 week CS exposed mice challenged with H1N1 for 48hrs (n=7-8) exposed WT and IL36 γ ^{-/-} mice. Depicted are mean values \pm SEM of biological replicates. (a, c-g) *P \leq 0.05, **P \leq 0.01, ***P \leq 0.001, ****P \leq 0.0001 vs all other groups by one-way ANOVA and Tukey's correction. (b) *P \leq 0.05, **P \leq 0.01, ***P \leq 0.001, ****P \leq 0.0001 vs untreated by t-test.

- The headlines in the results section are quite redundant. The authors should try to specify/sum up the results from each figure in the headline. In addition, there is often no summary of the results at the end of the results section.

We have changed headlines in the results section and have included a summary of the results in each results section.

- How long were the mice exposed to CS in fig 5 B, C, D, E, F, G? Please include this information in the figure legend.

We have included the duration of the CS exposure in Figure 5.

Fig. 5 IL-36 γ is critical in neutrophilic lung inflammation a Neutrophil counts in BALF from room air (RA) (n=18) exposed, or from 1 week (n=9), 2 week (n=9), and 3 week (n=8) cigarette smoke (CS) exposed mice, and from H1N1 (4 days post treatment) exposed mice (n=8), and from mice exposed to 2 weeks CS followed by 48hrs of H1N1 exposure (n=8). b Relative mRNA amounts of IL-36 γ in the cellular BAL pellet from room air (RA) and 2 week cigarette smoke (CS) exposed mice (line, n=number). c-d neutrophil numbers in BALF and IL-1 β , CXCL1, and IL-6 protein concentrations in lung homogenate of room air exposed mice (RA; n=4), 2 week CS exposed mice (n=4-8) and 2 week CS exposed mice challenged with H1N1 for 48hrs (CS+H1N1) (n=4-8). e Neutrophil numbers in BALF and relative mRNA amounts of IL-36 γ and Cxcr2 in lung homogenate from room air exposed (RA, n=5) and 2 week CS exposed mice challenged with H1N1 for 48hrs (n=7-8) exposed WT and IL36 γ ^{-/-} mice. f,g IL-1 β , CXCL1, IL-6 and TNF protein concentrations in BALF and in lung homogenate from room air exposed (RA, n=5) and 2 week CS exposed mice challenged with H1N1 for 48hrs (n=7-8) exposed WT and IL36 γ ^{-/-} mice. Depicted are mean values \pm SEM of biological replicates. (a, c-g) *P \leq 0.05, **P \leq 0.01, ***P \leq 0.001, ****P \leq 0.0001 vs all other groups by one-way ANOVA and Tukey's correction. (b) *P \leq 0.05, **P \leq 0.01, ***P \leq 0.001, ****P \leq 0.0001 vs untreated by t-test.

- Figure 3E: the stimulation conditions are not clear as well as the analyzed genes (two times IL36r?)

We have corrected the description of the stimulation condition.

Fig 3 IL-36y as an upstream amplifier in mouse macrophages and fibroblasts. a Relative mRNA amounts of *IL36r* in naive mouse bone marrow derived macrophages (BMDMs, n=6) in response to no stimulation (-) or stimulation with GM-CSF and IL-36y. **b** *CXCL1* protein concentrations in supernatant of BMDMs (n=4) after no stimulation (-), or stimulation with IL-36y, IL-1α, IL-1β alone or in combination with GM-CSF and TGFβ. **c,d** Relative mRNA amounts of *IL36g*, *Cxcl1*, *IfiA*, *IfiB*, *Ifi1*, *IfiM* and *IL36r* in BMDMs (n=4) (**c**) and primary mouse fibroblasts (n=4) (**d**) after no stimulation (-), or stimulation with IL-36y, IL-1α, IL-1β alone or in combination with GM-CSF and TGF-β. **e** Relative mRNA amounts in primary mouse fibroblasts (n=4) of *IL36r* after no stimulation (-) or stimulation with IL-36y, GM-CSF and TGF-β. Shown are the mean values ± SEM of biological replicates. (n=4) *P < 0.05, **P < 0.01, ***P < 0.001, ****P < 0.0001 vs all other groups by one-way ANOVA and Tukey's correction. (e) *P < 0.05, vs untreated by *t* test.

REVIEWERS' COMMENTS:

Reviewer #1 (Remarks to the Author):

All concerns were addressed successfully.

Reviewer #2 (Remarks to the Author):

I would like to thank the authors for the appreciation of my comments. The mentioned concerns were responded thoroughly and their work should be published.